# Kinetics and impacting factors of HO₂ uptake onto submicron atmospheric aerosols during a 2019 air quality study (AQUAS) in Yokohama, Japan

Jun Zhou[a,b,c]*, Kei Sato[d], Yu Bai[e], Yukiko Fukusaki[f], Yuka Kousa[f], Sathiyamurthi Ramasamy[d], Akinori Takami[d], Ayako Yoshino[d], Tomoki Nakayama[g], Yasuhiro Sadanaga[h], Yoshihiro Nakashima[i], Jiaru Li[c], Kentaro Murano[c], Nanase Kohno[c], Yosuke Sakamoto[c,d,e], Yoshizumi Kajii[c,d,e]*

[a] Institute for Environmental and Climate Research, Jinan University, 511443 Guangzhou, China

[b] Guangdong-Hongkong-Macau Joint Laboratory of Collaborative Innovation for Environmental Quality, Guangzhou 511443, China

[c] Graduate School of Global Environmental Studies, Kyoto University, Kyoto, 606-8501, Japan

[d] Center for Regional Environmental Research, National Institute for Environmental Studies, Tsukuba, Ibaraki 305-8506, Japan

[e] Graduate School of Human and Environmental Studies, Kyoto University, Kyoto 606-8501, Japan

[f] Yokohama Environmental Science Research Institute, Yokohama Kanagawa 221–0024, Japan

[g] Faculty of Environmental Science and Graduate School of Fisheries and Environmental Sciences, Nagasaki University, Nagasaki 852-8521, Japan

[h] Graduate School of Engineering, Osaka Prefecture University, Sakai, Osaka 599-8531, Japan

[i] Graduate School of Agriculture, Tokyo University of Agriculture and Technology, 3-5-8 Saiwai-cho, Fuchu, Tokyo 183-8538, Japan

*Corresponding author.

Graduate School of Global Environmental Studies, Kyoto University, Kyoto 606-8501, Japan

E-mail address: kajii.yoshizumi.7e@kyoto-u.ac.jp and junzhou@jnu.edu.cn

# Abstract

$HO_2$ uptake kinetics onto ambient aerosols play pivotal roles in tropospheric chemistry but are not fully understood. Field measurements of aerosol chemical and physical properties should be linked to molecular level kinetics; however, given that the $HO_2$ reactivity of ambient aerosols is low, traditional analytical techniques are unable to achieve this goal. We developed an online approach to precisely investigate the lower limit values of (i) the $HO_2$ reactivities of ambient gases and aerosols and (ii) $HO_2$ uptake coefficients onto ambient aerosols ($\gamma$) during 2019 air quality study (AQUAS) in Yokohama, Japan. We identified the effects of individual chemical components of ambient aerosols on $\gamma$. The results verified in laboratory studies on individual chemical components: transition metals play a key role in $HO_2$ uptake processes and chemical components indirectly influence such processes (i.e., through altering aerosol surface properties or providing active sites), with smaller particles tending to yield higher $\gamma$ values than larger particles owing to the limitation of gas phase diffusion is smaller with micrometer particles and the distribution of depleting species such as transition metal ions is mostly distributed in accumulation mode of aerosol. The modeling of $\gamma$ utilized transition metal chemistry derived by previous studies, further confirming our conclusion. However, owing to the high NO concentrations in Yokohama, peroxy radical loss onto submicron aerosols has a negligible impact on $O_3$ production rate and sensitivity regime.

# 1 Introduction

As an important atmospheric trace gas, the hydroperoxyl radical ($HO_2$) links many of the key oxidants in the troposphere, including the hydroxyl radical (OH), nitrate radical ($NO_3^-$), ozone ($O_3$), and hydrogen peroxide ($H_2O_2$) (Logan et al., 1981; Chen et al., 2001; Jaeglé et al., 2000; Sommariva et al., 2004; Jacob, 2000). However, the observed $HO_2$ concentration in field measurements has not yet been fully explained by sophisticated models (known as the "HOx dilemma") (Stone et al., 2012; Creasey et al., 1997; Kanaya et al., 2007b; Whalley et al., 2010; Millán et al., 2015), although it can be mostly solved in the conditions of clean marine air where NO concentration is low or aerosol loading is low

enough to make the heterogeneous reaction of HO$_2$ not important (Sommariva et al., 2004; Kanaya et
al., 2007a). Owing to the short atmospheric lifetime of HO$_X$(=OH+HO$_2$+RO$_2$), the HOx reactivity
measurement can provide a robust test of its complex chemistry (Heard and Pilling, 2003). The HO$_2$
uptake kinetics onto ambient aerosols, including HO$_2$ reactivity ($k_a$) and uptake coefficient ($\gamma$), influence
many atmospheric processes, including ozone formation rate, ozone formation sensitivity to NO$_X$, and
H$_2$O$_2$ formation (Sakamoto et al., 2019; Thornton et al., 2008). With $\gamma > 0.1$, HO$_2$ concentration can
also be influenced under conditions such as low [NO] or high aerosol loading (Lakey et al., 2015; Mao
et al., 2013b; Martínez et al., 2003; Tie et al., 2001, Jacob, 2000; Matthews et al., 2014). These effects
make the HO$_2$ uptake onto ambient aerosols indirectly influence human health and climate change.
From laboratory, field, and modeling studies, HO$_2$ uptake coefficients onto different types of
aerosol can span several orders of magnitude (~0.002–1), which can be affected by many parameters,
such as droplet/particle size and composition, the presence of dissolved reactive gases, and
environmental conditions (i.e., relative humidity (RH), pH, and $T$) (Taketani et al., 2012; Taketani et al.,
2008; Bedjanian et al., 2005; Thornton et al., 2008; George et al., 2013; Lakey et al., 2016a; Lakey et
al., 2016b; Matthews et al., 2014; Cooper and Abbatt, 1996; Hanson et al., 1992; Thornton and Abbatt,
2005; González Palacios et al., 2016; Mozurkewich et al., 1987; Remorov et al., 2002; Jaeglé et al.,
2000; Loukhovitskaya et al., 2009; Stone et al., 2012). In the absence of metals, the uptake of HO$_2$ by
ambient aerosols is believed to occur *via* the acid–base dissociation of HO$_2$ ( HO$_2$(g)
$\leftrightarrow$ HO$_2$(aq); HO$_2$ $\leftrightarrow$ O$_2^-$ + H$^+$, $pKa = 4.7$), followed by electron transfer from O$_2^-$ to HO$_2$ (aq) (HO$_2$ +
O$_2^-$ $\xrightarrow{\text{H}_2\text{O}}$ H$_2$O$_2$ + O$_2$ + OH$^-$), producing H$_2$O$_2$ (Jacob, 2000; Thornton et al., 2008; Zhou et al., 2019b).
However, laboratory studies have shown that certain transition metals (i.e., Cu(II) and Fe(II)) can act
as catalysts and accelerate HO$_2$ uptake rates onto many chemical compounds (Thornton et al., 2008;
Taketani et al., 2008; Taketani et al., 2012, Cooper and Abbatt, 1996). Owing to the sufficiently high
metal concentrations in tropospheric aerosols, as shown in previous field measurements (Hofmann et
al., 1991; Wilkinson et al., 1997; Guieu et al., 1997; Manoj et al., 2000; Halstead et al., 2000; Siefert et
al., 1998; Sedlak and Hoigné, 1993; Guo et al., 2014), recent studies have proposed that $\gamma$ may be
dominated by metals contained in the aerosol. This can lead to the HO$_2$ destruction (Mao et al., 2013a;
George et al., 2013), forming $H_2O_2$, $HO_2$–water complexes, or water and sulfate (Mozurkewich et al.,
1987; Cooper and Abbatt, 1996; Gonzalez et al., 2010; Loukhovitskaya et al., 2009; Mao et al., 2010;
Macintyre and Evans, 2011), which are important in the evolution of the chemical composition and
physical properties of particles (George and Abbatt, 2010; George et al., 2008). The available data
concerning $HO_2$ uptake kinetics onto ambient aerosols are insufficient for quantitative consideration
owing to the much lower $k_a$ value, as compared with the $HO_2$ reactivity of ambient gases ($k_g$). Therefore,
they are below the detection limits of the current instruments.

87         To our knowledge, aside from us, only one study has measured $\gamma$, using an offline method that

integrated ambient aerosols over size and time (Taketani et al., 2012). Considering that the offline
method may distort $\gamma$, we developed an online approach to evaluate $HO_2$ uptake kinetics onto ambient
aerosols. This method was successfully applied in Kyoto, Japan, in summer 2018, using a versatile
aerosol concentration enrichment system (VACES) and a technique combining laser-flash photolysis
with laser-induced fluorescence (LFP–LIF) (Zhou et al., 2019b). The obtained average $\gamma$ value (~0.24)
was comparable with the previous values used for modeling studies (~0.2) (Stadtler et al., 2018; Jacob,
2000). However, the large standard deviation ($\pm$ 0.20, 1 $\sigma$) of $\gamma$ along with the measurement time suggest
that many other parameters might play a role, e.g., the measurement setup, aerosol characteristics, $T$,
and RH.

97         In this study, we chose Yokohama (Japan), a coastal city with higher pollutant levels than Kyoto

and different properties of the air masses from mainland Japan and the coast, as the measurement site.
This is part of the Air QUAlity Study (AQUAS) campaigns. The chemical and physical properties of
ambient aerosols were quantified in real-time. To test their influence on $k_a$ and $\gamma$, we conducted
correlation matrix analysis coupled with the bootstrap method and classified the arriving air masses
from different directions. Further, the main mechanism of $\gamma$ was investigated by comparing the real-
time quantified $\gamma$ values with the modeled values. The impact of the peroxy radical's loss onto ambient
aerosols on air quality is evaluated through their impact on ozone formation rates and their sensitivity
to $NO_X$. The results obtained here will for better estimation of the heterogeneous reaction between $HO_2$
and ambient aerosols in sophisticated air quality models.

# 2 Materials and methods

**2.1 Sampling sites**

The measurement campaign was conducted at Yokohama Environmental Science Research Institute in Yokohama, Japan (location: 35°28'52.8"N, 139°39'30.3"E), from July 24 to August 03, 2019. The sampling ports of the instruments were placed approximately 25 m above the ground. Figure S1 shows the air mass directions during the campaign, which can be classified into two categories: (i) from the sea to the north, toward Yokohama City (~19% of the experimental period: from 12:00 July 25 to 12:00 July 27, 2019) and (ii) from the mainland toward Yokohama City (~81% of the experimental period). This classification was intended to distinguish the chemical properties of aerosols arriving from the mainland and the ocean, and to consequently quantify their impacts on $k_a$ and $\gamma$.

**2.2 Measurement strategies, instrumentation, and related data analysis**

*LFP–LIF* In situ ambient air $HO_2$ reactivity was measured using LFP–LIF, which was adapted from a laser-induced pump and probe OH reactivity measurement technique. This approach has been successfully employed for gas and aerosol phase HOx (=OH+$HO_2$) reactivity measurements (Sadanaga et al., 2004; Miyazaki et al., 2013; Sakamoto et al., 2018). Further details concerning the $HO_2$ reactivity measurements are described in the Supporting Information (SI).

*VACES* To compensate for the relatively low ambient aerosol concentrations thus the low $k_a$, and the low limit of detection (LOD) for the $HO_2$ reactivity measurement (~0.003 s$^{-1}$ with 240 decay integrations), a setup with VACES and an auto-switching aerosol filter was used before LFP–LIF. The VACES was built according to Sioutas et al. (1999), the ambient air sample was drawn into a tank (containing ultra-pure water heated up to ~32 °C) of VACES through a PM$_{2.5}$ cyclone at a flow rate of over ~ 100 L min$^{-1}$, where the ambient air steam was saturated and subsequently cooled down in a condenser connected above the tank (with a temperature of $-2$ °C). During this process, the water droplets with diameters >2 μm formed on the collected ambient aerosols, which were then enriched by a virtual impactor (with a 50% cutoff point less than 1 μm) and dried by passing through a diffusion dryer connected right after the condenser in sequence. The concentration enrichment of the ambient

aerosols can be estimated using the total intake flow of VACES and the minor output flow of the virtual
impactor that connected to the aerosol instrumentations (more details are given in SI: the enrichment of
the ambient aerosols). Wang et al. (2013, 2014) claimed that when using the same technique as VACES
for the online measurement of copper in ambient aerosols, equivalent copper concentrations were
obtained compared to those measured by inductively coupled plasma mass spectrometer (ICP–MS) for
both total and water-soluble components, which indicates the impact of VACES system to the solubility
of Cu contained in ambient aerosol is negligible. Furthermore, previous studies found the liquid-liquid
phase separation RH ranged from 60%-100% in atmospherically relevant particles consisting of organic
species and inorganic salts (Yu et al., 2014), and the organic component appears to be the most useful
parameter for estimating the liquid–liquid phase separation, which was always observed for oxygen-to-
carbon elemental ratio (O:C) < 0.5 and was never observed for O:C $\geq$ 0.8 (Bertram et al., 2011). In this
study, the ambient aerosol O:C ranged from 0.1 to 0.7, and the RH changing from ~80% (in ambient
air) to >100% (in water tank), and then to ~75%(in reaction cell), suggest that the phase separation may
have already happened before entering the VACES system, thus we assume the morphology of the
ambient aerosols didn't change during the concentration enrichment process. Unfortunately, we did not
measure the chemical composition after the VACES, thus we are not able to compare the chemical
composition of the post VACES aerosols to ambient aerosol. However, previous test using the ambient
aerosol fractions including coarse and fine PM indicated that the enrichment process of the VACES system
does not differentially affect the chemical composition of ambient PM (Kim et al., 2001), thus we assume
the chemical composition changing due to the enrichment process of the VACES can be neglected. The
enrichment factor of the ambient aerosol surface area (*E*) was calculated from the difference between
the surface areas measured before and after VACES by two scanning mobility particle sizers (SMPSs).
***Aerosol physical properties and the enrichment factor of VACES*** The mass concentration and surface
area of ambient aerosols (before VACES) were determined using a Scanning Mobility Particle Sizer
(SMPS$_1$, model 3936L72, TSI, measure particle size distribution at 14.1–736.5 nm, 5-min intervals).
The mass concentration of PM$_{2.5}$ was measured using a palm-sized optical PM$_{2.5}$ sensor (Nakayama et
al., 2018). In order to test the enrichment factor of the VACES, a SMPS$_2$ (model 3936L75, TSI, measure
particle size distribution at 14.6–661.2 nm, 5-min intervals) was used to measure the enriched mass
concentration and surface area of ambient aerosols (after VACES) for ~ 2 hours every day for ~ 6 days.
The enrichment factor of VACES for the surface area was estimated as $12.5 \pm 2.5$ from the ratio
between $S_2$ and $S_1$, where $S_2$ and $S_1$ are the averaged surface areas measured by $SMPS_2$ and $SMPS_1$ of
each day, respectively. According to the test from previous study of the VACES system, there is no
distortion of the size distribution of the original ultrafine aerosols as the particle concentration
enrichment occurs without any coagulation (Sioutas et al., 1999), here we listed the mean radius and
geometric standard deviation (Geo. Std. Dev.) of the ambient aerosols before and after VACES during
the enrichment factor measurement periods, as shown in Table 1. We could see that the mean radius
before and after VACES are not statistically different within the standard deviation.
**Table 1**: The Mean radius and Geometric standard deviation of ambient aerosols before and after VACES

| Experimental time• | Before VACES | | After VACES | |
|---|---|---|---|---|
| | Mean radius (nm) | Geo. Std. Dev. | Mean radius (nm) | Geo. Std. Dev. |
| 2019.7.25 09:03 − 11:03 | 129.47±11.32 | 0.92±0.04 | 133.19±3.37 | 0.92±0.02 |
| 2019.7.26 09:30 − 11:30 | 94.95±14.42 | 0.99±0.09 | 85.09±14.96 | 1.01±0.09 |
| 2019.7.27 10:00 − 12:00 | 85.09±14.96 | 1.01±0.09 | 80.40±21.01 | 1.01±0.07 |
| 2019.7.28 09:30 − 11:30 | 163.62±13.32 | 1.01±0.08 | 164.06±14.40 | 1.04±0.06 |
| 2019.7.29 09:10 − 11:10 | 128.06±6.90 | 0.91±0.02 | 125.07±7.68 | 0.92±0.02 |
| 2019.7.30 09:30 − 11:30 | 111.40±8.21 | 1.01±0.02 | 115.32±6.26 | 1.01±0.03 |

•represent the time period of the enrichment factor measurements;
±represent the standard deviation of the averaged values of mean radius and Geo. Std. Dev.
The enriched surface area of ambient aerosols with aerodynamic diameter < 0.74 μm ($PM_{0.74}$) was
calculated from the surface area of ambient aerosol measured by $SMPS_1$ and the enrichment factor. The
enriched surface area of $PM_{2.5}$ was then calculated by multiplying the enriched surface area of $PM_{0.74}$
by the mass ratio between $PM_{2.5}$ and $PM_{0.75}$ (~1.1), where we assume the surface area are increased in
proportional to the mass concentration. However, as the larger particles (here referred to particles ranged
from 0.74 to 2.5 μm) tend to have lower surface area than the smaller particles, we consider the obtained
enriched surface area of $PM_{2.5}$ as the upper limit value. More details can be found in SI.
***HO$_2$ uptake kinetics*** After passing through the VACES system, the ambient air was sampled using a
three-port valve (Bolt, Flon Industry Co., LTD) and injected into the LFP–LIF system. The valve was
switched automatically between two sampling lines, one with the aerosol filter on, and the other one
with the aerosol filter off, HO$_2$ reactivities in ambient air caused by two modes were measured: (a) the
gas phase mode with aerosol filter on, the HO$_2$ reactivities are represented as $k_g$, and (b) the gas +
enriched aerosol phase mode with aerosol filter off, the HO$_2$ reactivities are represented as $k_g+Ek_a$, where
$E$ represents the enrichment factor of $k_a$, $Ek_a$ represents the total HO$_2$ reactivities caused by enriched
ambient aerosols, the usage of $Ek_a$ is based on the assumption that HO$_2$ uptake with aerosol particles
follows the pseudo-first-order rate law. We modeled $k_g$ in both modes using a theory identified
previously (see SI: HO$_2$ reactivity of ambient gas phase) (Zhou et al., 2019b) and compared it with the
measured values. The differences between measured and modeled $k_g$ in mode (a) enabled us to establish
their interrelationship and to check instrument stability. The differences between ($k_g+Ek_a$) and the
modeled $k_g$ in mode (b) are considered as the enriched aerosol phase HO$_2$ reactivity ($Ek_a$). The total HO$_2$
reactivity decay profile follows single-exponential decay:
$$HO_2 = [HO_2]_0 \exp\left(-(k_g + Ek_a + k_{bg})t\right) \tag{1}$$
where $k_{bg}$ denotes the zero air background obtained by injecting zero air with the same RH as the real-
time ambient value into the reaction cell every 24 h for 30 min. The RH was controlled by passing some
of the zero air through a water bubbler. The value of $k_{bg}$ was subtracted separately on each day. The
variability of $k_{bg}$ (i.e., the reproducibility of the laser system) was calculated as the standard deviation
of the response of repeated measurements on different days. It was found to be ~4%, which is slightly
higher than the instrument precision (3%). A 30-min average calculation was applied to the data to
reduce data fluctuation. The observed HO$_2$ uptake coefficients onto ambient aerosols ($\gamma_{obs}$) can be
calculated from the dependence of $Ek_a$ on $\gamma_{obs}$:
$$Ek_a = \frac{\gamma_{obs}\omega_{HO_2}ES}{4} \tag{2}$$
where $ES$ and $\omega_{HO_2}$ represent the enriched surface area of ambient aerosol after VACES and the mean
thermal velocity of HO$_2$ (~437.4 m s$^{-1}$), respectively. The uncertainty of the enriched surface area was
estimated from the instrument systematic error of SMPS (~ 8%) and the uncertainty of the enrichment
factor ($\pm2.5$), which are shown in Fig.1b (see SI). The HO$_2$ reactivity of ambient aerosol ($k_a$) can be
obtained from $Ek_a$ by dividing by the enrichment factor $E$.
***High resolution–time of flight–aerosol mass spectrometry (HR–ToF–AMS)*** A field-deployable HR–
ToF–AMS (Aerodyne Research Inc.) (DeCarlo et al., 2006) was used for the characterization of the
non-refractory aerosol mass with a time resolution of ~3 min. The HR-ToF-AMS measured the total
organic aerosol (OA), sulfate ($SO_4^{2-}$), nitrate ($NO_3^-$), ammonium ($NH_4^+$), chloride ($Cl^-$), and the two
most dominant oxygen-containing ions in the OA spectra, i.e., mass-to-charge ratios of m/z = 44 (Org44,
mostly $CO_2^+$) and m/z = 43 (Org43, mainly $C_2H_3O^+$ for the oxygenated OA and $C_3H_7^+$ for the
hydrocarbon-like OA) (Ng et al., 2011). The fractions of Org44 and Org43 in OA are represented as $f_{44}$
and $f_{43}$, respectively. Ambient air was sampled through a critical orifice into an aerodynamic lens, which
efficiently transmitted particles between 80 nm and up to at least 1 μm. Particles were flash-vaporized
by impaction on a resistively heated surface (~600 °C) and ionized by electron ionization (70 eV). The
m/z values of the resulting fragments were determined using a ToF mass spectrometer. Data were
analyzed using the ToF–AMS software SQUIRREL and PIKA. Data were not corrected for lens
transmission efficiency. Standard relative ionization efficiencies (RIE) were used for organics (RIE =
1.4), nitrate (RIE = 1.1), chloride (RIE = 1.3), sulfate (RIE = 1.12), and ammonium (RIE = 4).
Concentration data were obtained from background-subtracted stick-mass data (low-mass-resolution-
base mass concentration data, which are calibrated using ammonium sulfate particles) and determined
assuming a collection efficiency (CE) of 0.5.
***Filter-based photometer*** Real-time measurement of the equivalent black carbon (eBC) was performed
using a 5-wavelength dual-spot absorption photometer (MA300, AethLabs, San Francisco, CA, USA),
which performed an online correction for possible artefacts resulting from filter loading and multiple
scattering (Drinovec et al., 2015). In this study, eBC data obtained from light attenuation at a wavelength
of 880 nm were used to avoid possible contributions from brown carbon; the time resolution was ~1
min.
***Trace elements*** Fourteen trace elements (Al, V, Cr, Mn, Co, Ni, Cu, Zn, As, Se, Sr, Cd, Ba, and Pb)
were measured using an offline method at two-day intervals from 21 July to 5 August 2019. The
suspended particulate matter (SPM) was collected onto 623.7 cm$^2$ size quartz fiber filters (Pallflex
Tissuquartz 2500QAT-UP), which had an available collecting area of 405.84 cm$^2$, using a high-volume
sampler (1000 L min$^{-1}$). Approximately 2 cm$^2$ of each filter was cut into pieces and placed into a
polytetrafluoroethylene (PTFE) pressure digestion tank with 1 mL 49% hydrofluoric acid (HF) and 5
mL 69% nitric acid (HNO$_3$). A Thermo Fisher X2 Series ICP–MS was then used to determine metal
concentrations. By assuming that the metal fractions were the same in SPM and PM$_1$ (aerosol
particles with aero-dynamic diameters less than 1 µm), the concentrations in PM$_1$ were estimated
according to the tested metal concentrations in SPM and the ratio between SPM and PM$_1$ measured in-
situ.
***Water-soluble inorganic species*** NR-PM$_1$ water-soluble inorganic species (including Na$^+$, SO$_4^{2-}$, NH$_4^+$,
NO$_3^-$, Cl$^-$, Ca$^{2+}$, K$^+$, Mg$^{2+}$) used for *the ISORROPIA-II model* were also measured using offline method,
as described above. For extraction, we cut 1/4 of a 47 mm filter punched from the original collected
filter and placed it in 10 mL of ultrapure water (18.2 MW cm$^{-1}$) in a centrifuge tube. This was followed
by 15 min of ultrasonication in a 30$^o$C water bath. The solution was then vortexed (Vortex Genie 2,
Scientific Industries, USA) for 1 min to ensure homogeneity and filtered through syringe filter with
pore size of 0.45-µm (Advantec Dismic-25, PTFE). An Ion Chromatograph (IC, ICS1600, DIONEX,
USA) was employed to determine the concentrations of these inorganic ions in the extracted solution.
***Gas phase monitors*** NO$_2$ was measured by cavity attenuated phase shift (CAPS, Aerodyne Research,
USA, at 1-s intervals), NOy－NO by chemiluminescence (Model 42i-TL, Thermo, at 10-s intervals),
CO by Thermo CO analyzer of nondispersive infrared spectroscopy (Model 48i-TLE, Thermo Scientific,
USA, at 10-s time intervals), and O$_3$ by UV absorption (Model 1150, Dylec, AMI Co., Ltd, at 10-s time
intervals). HCHO was determined by high performance liquid chromatography (HPLC; 1260 Infinity,
Agilent Technologies Inc, USA) from 14:00 July 29, to 12:00 August 3, 2019. An average value of ~2
ppb was used for the data analysis.
***ISORROPIA-II model*** NR-PM$_1$ water-soluble inorganic species (including Na$^+$, SO$_4^{2-}$, NH$_4^+$, NO$_3^-$, Cl
$^-$, Ca$^{2+}$, K$^+$, Mg$^{2+}$) and meteorological parameters including temperature and RH were used to calculate
the aerosol pH and liquid water content based on the *ISORROPIA-II* model (Fountoukis and Nenes,
2007). We ran *ISORROPIA-II* in "reverse" mode and the particles were assumed to be deliquescent, i.e.,
in metastable mode (Hennigan et al., 2015). The thermodynamic equilibrium of the $NH_4^+$- $SO_4^{2-}$- $NO_3$
$^-$ system case was used for modeling.

# 3 Results and discussion

**3.1 The HO₂ uptake kinetics onto ambient aerosols**

The measured total $HO_2$ reactivities were compared against the modeled gas phase $HO_2$ reactivity under
the experimental conditions both with and without the aerosol phase. Without the aerosol phase, the
modeled $k_g$ values are calculated according to the description in Sect. 2.2, which are not statistically
different with the measured $k_g$ values (Fig. 1a second panel, T-test, $p = 0.49$, with inspection level =
0.05), indicating that $HO_2$ loss in the reaction cell was dominated by its reaction with $NO_2$ in the LFP-
LIF system. With the aerosol phase, the measured ($Ek_a+k_g$) and modeled values ($\approx k_g$) were significantly
different (see Fig. 2b, first panel, T-test, $p = 0.04$, with inspection level = 0.05). The differences were
considered to be the $HO_2$ reactivities of enriched ambient aerosols ($Ek_a$). $Ek_a$ ranged between 0.015 s$^{-1}$
(25$^{th}$ percentile) and 0.097 s$^{-1}$ (75$^{th}$ percentile), with the median value of 0.059 s$^{-1}$, the corresponding $k_a$,
calculated by dividing $Ek_a$ by $E$, ranged between 0.001 s$^{-1}$ (25$^{th}$ percentile) and 0.008 s$^{-1}$ (75$^{th}$ percentile),
with the median value of 0.005 s$^{-1}$ and average value of 0.005 $\pm$ 0.005 s$^{-1}$. The error for $Ek_a$ was estimated
as ~ 0.05 s$^{-1}$, calculated as the propagated error from $k_g+Ek_a$ (i.e., the systematic error of the instrument,
~0.05 s$^{-1}$) and the modeled $k_g$ in mode (b) (~ 0.001 s$^{-1}$). The error for $k_a$ was then estimated as ~ 0.004
s$^{-1}$ by dividing the error of $Ek_a$ by the enrichment factor $E$. The corresponding $\gamma$, calculated from Eq.
2, ranged from 0.05 (25$^{th}$ percentile) to 0.33 (75$^{th}$ percentile), with the median value of 0.19 (with an
average value of 0.23 $\pm$ 0.21). The mean diameter of ambient particles ranged from 0.1 to 0.46 μm (with
the median value of 0.25 μm), the gas-phase diffusion effects on $\gamma$ were estimated to be ~ 6.6 % (further
details are given in the SI). The absolute increase of $\gamma$ due to the gas-phase diffusion is 0.03 on average,
which is negligible compared to $\gamma$ uncertainty (~0.21 on average). Therefore, we ignored the gas-phase
diffusion effects to $\gamma$.







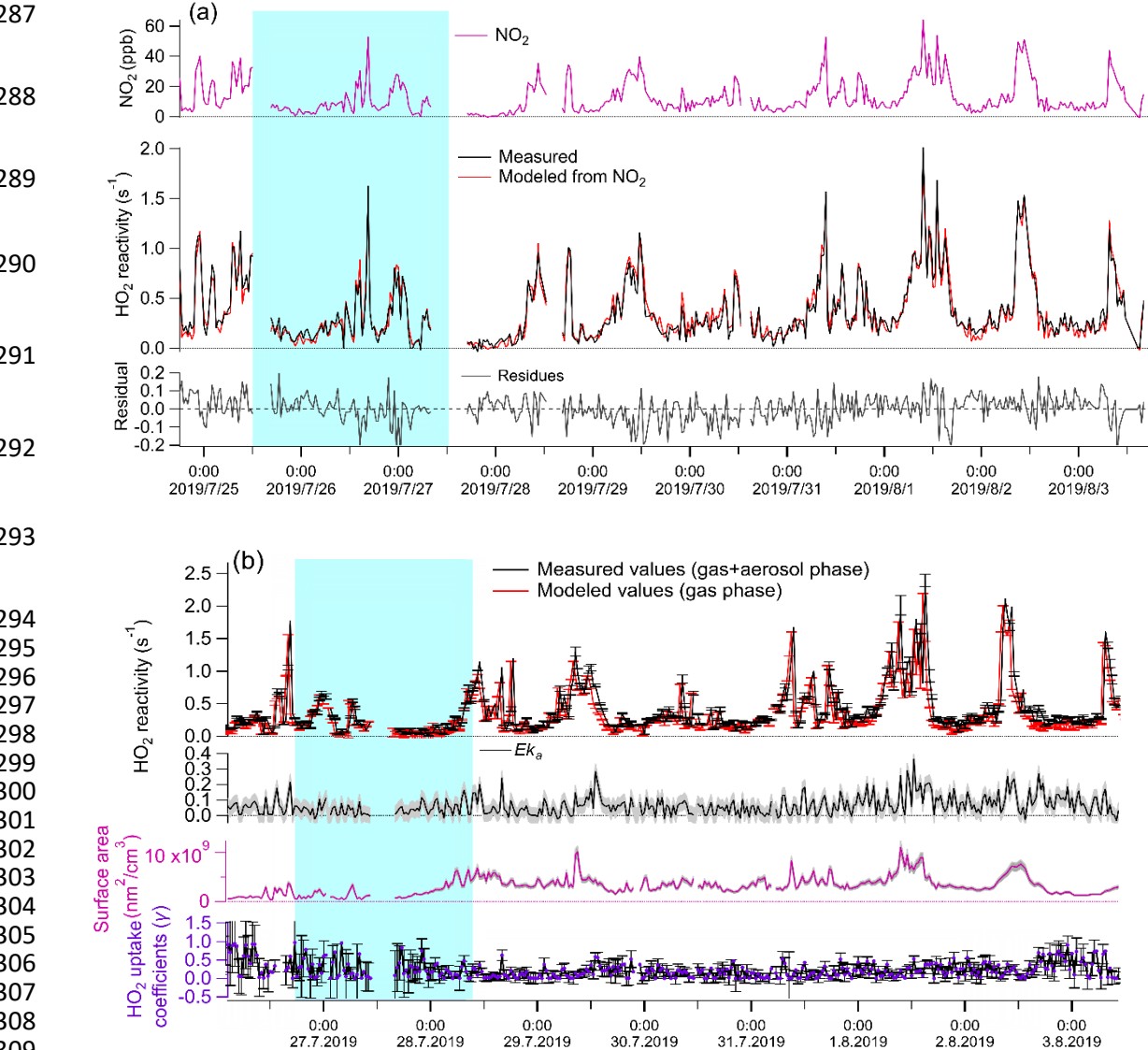



















Figure 1: Temporal variation of parameters under different experimental conditions. (a) Without aerosol phase:
1st panel: measured $NO_2$ concentrations (ppb); 2nd panel: measured (red line) and modeled (black line) $k_g$; 3rd panel:
fitting residues of modeled $k_g$ values, ranging from −0.04 (25 percentile) to 0.05 (75 percentile), therefore we
consider the systematic error of the LFP–LIF instrument to be ~0.05 s⁻¹. (b) Gas + aerosol phase: 1st panel:
measured total $HO_2$ reactivity ($k_g+Ek_a$) and modeled $k_g$; 2nd panel: $Ek_a$, calculated from the difference between the
measured and modeled values from the 1st panel, the gray shadow area represents the uncertainty of $Ek_a$ ($\Delta Ek_a$),
propagated from the error of ($k_g+Ek_a$) and modeled $k_g$; 3rd panel: the upper limit surface area of the enriched
ambient aerosols ($ES$), the gray shadow area represents the uncertainty of $ES$ ($\Delta ES$), propagated from the
systematic errors of the SMPS instrument (~8%), and the uncertainty of the enrichment factor; 4th panel: $\gamma$
calculated from $Ek_a$ and $ES$ according to Eq. 2. The errors for $\gamma$ were propagated from $\Delta Ek_a$ and $\Delta ES$, $\Delta\gamma = \gamma$
$\times \sqrt{\dfrac{\Delta Eka^2}{Eka} + \dfrac{\Delta ES^2}{ES}}$ . The blue shaded area represents the air masses from group i (from coast), the remainder is from
group ii (from mainland).

324        Statistical significance analysis showed that the average $\gamma$ value of group i (0.35 ± 0.28) is

significantly higher than that of group ii (0.21 ± 0.16) (calculated $p$ = 4.9E-5; Mann-Whitney),
indicating that the air masses from the ocean yield higher $\gamma$ values than the air masses from mainland
Japan. The difference in $\gamma$ values between group i and group ii may due to the different chemical
components contained in the ambient aerosols arrived from the ocean or mainland, which we will
discuss in the following sections. The average value of $k_a$ at Yokohama ($0.005 \pm 0.005$ s$^{-1}$) was much
higher than that at Kyoto city ($0.0017 \pm 0.0015$ s$^{-1}$) (with calculated $p < 0.05$; Mann-Whitney), this may
due to the different aerosol properties in Kyoto and Yokohama city. We list some of them as follows:
1) mass composition, the aerosols at the coast city (Yokohama) tend to contain more sea salts thus
increased $k_a$, 2) particle size distribution, smaller particles tend to yield higher $\gamma$ values than larger
particles owing to the depleting species (e.g., transition metal ions) are mostly distributed in
accumulation mode of aerosol, 3) the water content and the metal concentrations, which will highly
influence the HO$_2$ uptake capacity of the ambient aerosols. However, the average value of HO$_2$ uptake
coefficient onto ambient aerosols ($\gamma$) at Yokohama is ~0.23, which is comparable with previous
measured (~0.24–0.25) (Zhou et al., 2019b; Taketani et al., 2012) and modeled (~0.20) values (Stadtler
et al., 2018; Jacob, 2000). The large standard deviation ($\pm$ 0.21, $1\sigma$) of the values along with the
measurement time may be due to the instantaneously changed chemical and physical properties of
ambient aerosols, indicating that a large bias may exist if a constant $\gamma$ value is used for modeling.
**3.2 Bulk chemical composition of ambient aerosols**

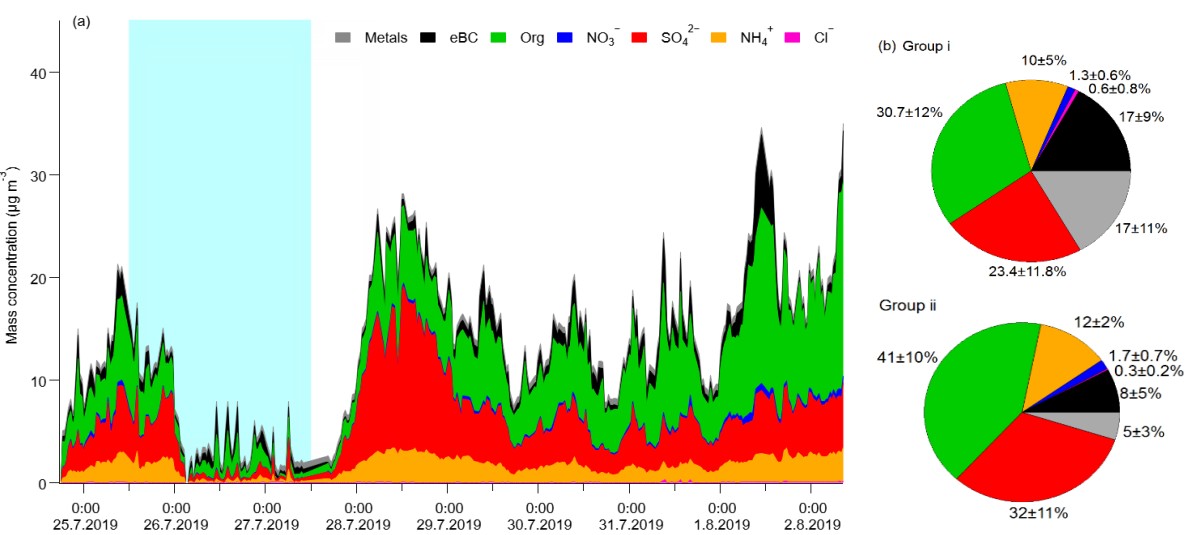

Figure 2: (a) Concentrations of non-refractory chemical components plus eBC in Yokohama, Japan (July 24 to
August 02, 2019). The blue shaded area represents group i from coast and the remaining areas represent group ii
from mainland. (b) Average contribution fractions of different chemical components of groups i and ii.
Figure 2a shows the time series of the mass concentrations of OA, SO$_4^{2-}$, NO$_3^-$, NH$_4^+$, Cl$^-$, and eBC in
PM$_1$ in Yokohama from July 24 to August 02, 2019, which is ~1.5 days less than for the LFP–LIF data.
During this period, PM$_1$ ranged from ~1 to 35 μ g m$^{-3}$ (average ≈ 13 μg m$^{-3}$) and was dominated by OA,
SO$_4^{2-}$, and NH$_4^+$, with contributions of 39 ± 11%, 30 ± 12%, and 12 ± 4%, respectively; these were
followed by eBC and metals, with contributions of 10 ± 7% and 8 ± 8%, respectively. Cl$^-$ contributed
< 1% in both groups, which is similar to that reported for an urban area in winter in Bern (Switzerland)
(Zhou et al., 2019a). However, NO$_3^-$ contributed much less (~2 ± 0.7%) compared with that reported
for Bern (~19 ± 4%), which may be due to the reverse reaction of NH$_4$NO$_3$ converting to HNO$_3$. Since
Yokohama is a coastal city, and HNO$_3$ is easy vaporized in summer, gaseous HNO$_3$ may sink with sea
salt particles by forming NaNO$_3$ through heterogeneous reactions (Finlayson-Pitts and Pitts, 2000).

358        Figure 2b shows the average contribution fractions of different components of group i and group

ii. The main differences in the components between these two groups are the fractions of OA, BC, SO$_4^{2-}$,
and metals. The OA fraction was ~1.8 and ~8.4 times higher than that for the metals in groups i and ii,
respectively. As OA can cover the surface of the particles and thereby decrease $\gamma$ (Lakey et al., 2016a;
Takami et al., 2013), the difference between the OA and metal fractions in these two groups may
partially explain the much higher $\gamma$ values of group i (vs. group ii). Previous studies have shown quite
low HO$_2$ uptake coefficient on BC (~ 0.01) (Saathoff et al., 2001; Macintyre and Evans, 2011), which
is different with the result obtained here. This may due to the much higher fraction of BC in group i
(vs. group ii) provide active sites for HO$_2$ self-reaction or its reaction with the H atom from the
abstraction reaction from hydrogen containing functional groups and form H$_2$O$_2$ (Bedjanian et al., 2005),
or BC can be coated with additional materials (e.g., sulfate and organic carbon) thus influence HO$_2$
uptake (Schwarz et al., 2008). We also observed slightly higher Cl$^-$ and BC fraction in group i (from
ocean) than that in group ii (from mainland), which may due to the effects of sea salt and the ship
emissions in the air mass from the ocean, respectively. From the average diurnal patterns (Figs. S5 and
S6), the trends in $k_a$ follow the trends in chemical composition, whereas $\gamma$ shows a contrasting trend
with both variables in both groups. For group ii, SO$_4^{2-}$ and OA exhibited higher values whereas $\gamma$
exhibited lower values during the daytime than those during nighttime, indicating that secondary aerosol
formation resulting from photochemical reactions may decrease $\gamma$. To identify the influence of each
individual chemical component of ambient aerosol on $k_a$ and $\gamma$, we further performed correlation matrix
analysis.
**3.3 Influence of individual chemical components of ambient aerosol on $k_a$ and $\gamma$**
For multiple−component ambient aerosol, $k_a$ and $\gamma$ are influenced by different chemical components,
these chemical components may also have mutual effects to each other, for example, the transition metal
Cu and Fe contained in ambient aerosols can be chelated by organics (Lakey et al., 2016b). Therefore,
we produced a Pearson correlation matrix of all the testing factors at Yokohama city, including different
chemical components, $k_a$ and $\gamma$. Here we note that the different chemical components were measured
using HR–ToF–AMS for ambient aerosols with aerodynamic diameters < 1 μm, while $k_a$ and $\gamma$ were
measured using VACES–LFP–LIF system for ambient aerosols with aerodynamic diameters < 2.5 μm,
but due to most "fine-mode" aerosols have the mean diameter ranged from 0.09 μm to 0.47 μm (with
the median value of 0.25 μm, measured by SMPS), we assume the chemical components of ambient
aerosols with the aerodynamic diameter ranged between 1 μm and 2.5 μm have negligible impact on
Pearson correlation matrix result. However, present results do not include the effects of coarse particles
(with aerodynamic diameters > 2.5 μm) to the $HO_2$ uptake kinetics, and we may partially miss
measuring sea spray (with diameters ranged from ∼ 0.05 to 10 μm) effects. When $Cl^-$ measured by
AMS increased, coarse particles may exist and our results may not represent the real ambient
conditions. Consequently, we consider our results as the lower limit of the $HO_2$ uptake kinetics onto
real ambient aerosols.

395        To exclude the effects of the different fractions of chemical components in groups i and ii, the

bootstrap method, which is based on the creation of replicate the inputs by perturbing the original data
through resampling, was employed. The resampling was performed by randomly reorganizing the rows
of the original time series such that some rows of the original data were present several times while
other rows were removed. The final results were obtained by running the data for 1000 bootstrap
replicates. The average values of these 1000 bootstrap replicates are listed in Fig. 3.

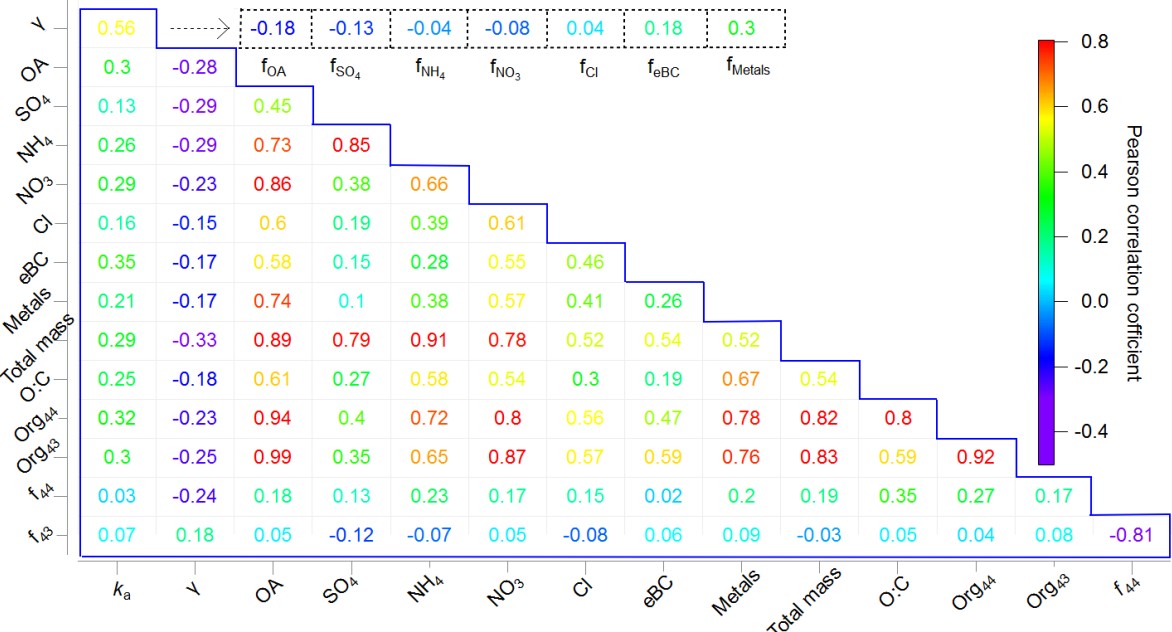

Figure 3: Correlation matrix showing Pearson's r values for the chemical compositions, $k_a$, and $\gamma$ during the corresponding measurement periods (in the blue box), as well as the Pearson's r values for the chemical composition fraction $i$ (represented as $fi$, $i$ = OA, $SO_4^{2-}$, $NH_4^+$, $NO_3^-$, $Cl^-$, eBC, and metals) and $\gamma$ (in the dashed line box).

Most of the chemical components had strong or moderate Pearson correlation coefficients with each other (Fig. 3), although $k_a$ and $\gamma$ showed only a moderate correlation with each other (0.56). As $\gamma$ can be correlated with the qualitative, rather than quantitative, properties of the aerosols, we further investigated the Pearson's r values between the chemical composition fractions and $\gamma$. The results show that $k_a$ was positively correlated with total mass and the individual chemical components, whereas $\gamma$ showed only a weak positive correlation with $f$metals (0.30) and $f_{eBC}$ (0.18). According to previous studies, metals may act as a catalyst thus accelerating the depletion of $HO_2$ (Mao et al., 2013a), and BC can provide active sites or can be coated by other chemical components thus facilitating the $HO_2$ uptake (Bedjanian et al., 2005; Schwarz et al. (2008), as described in Sect.3.2. The very weak correlation of $\gamma$ with $f_{Cl^-}$ (0.04) may be related to $Cl^-$ chemistry, for example, $HO_2(g)$ can react with NaCl(g), produce NaOH and $Cl_2(g)$, thus cause a decrease in the $HO_2$ concentration and indirectly effects $\gamma$ (Remorov et al., 2002). Interestingly, when considering the Org44 and Org43 fractions in total OA separately, $\gamma$ is positively correlated with $f_{43}$ (0.18) but negatively correlated with $f_{44}$ (−0.24). This is consistent with previous conclusion that more oxidized organic aerosols tend to be highly viscous and thus decrease $HO_2$ uptake coefficients (Lakey et al., 2016b). In summary, $\gamma$ was dominated by the free forms of

transition metals that can act as catalysts of $HO_2$ uptake onto ambient aerosols, and was indirectly
affected by chemical components that might alter the properties of ambient aerosols, e.g., oxygenated
OA can cover the aerosol surface and alter the viscosity of ambient aerosols, thereby decrease $\gamma$(Lakey
et al., 2016a; Lakey et al., 2016b; Takami et al., 2013), whereas BC may provide active sites or be
coated by other chemical components, thereby increase $\gamma$. This is further confirmed by the classification
of the air masses, i.e., the air mass from the ocean (group i), which contained less OA and more metals
than that from mainland Japan (group ii), had a higher $HO_2$ uptake capacity. We further compared the
measured $\gamma$ values with the modeled $\gamma$ values using previously proposed mechanisms, as shown below.
**3.4 Possible mechanism of $HO_2$ uptake onto ambient aerosols**
Two mechanisms of $HO_2$ uptake onto aqueous ambient aerosols have been proposed, for which
equations have been derived from a previous study (Thornton et al., 2008): (i) $HO_2$-only chemistry and
(ii) chemistry with transition metals playing a role. In this study, the liquid content of the total ambient
aerosol mass ranged from 70% to 88%, as obtained from the *ISORROPIA-II model*. As the solubility of
Fe is rather small in ambient aerosol, the reaction rates of Fe/Mn for liquid phase $HO_2$ in aerosol is ~
100 times slower than it is for Cu, thus the influence of Fe and Mn on $HO_2$ uptake can be neglected
compared to Cu or scaled as equivalent [$Cu^{2+}$] (Fang et al., 2017; Hsu et al., 2010; Baker and Jickells,
2006; Oakes et al., 2012; Song et al., 2020), therefore, we use the soluble Cu as surrogate for transition
metals in ambient aerosols to assess their influence to $\gamma$. The Cu solute mass fraction in the liquid content
of the ambient aerosols was estimated as $(3.5–30) \times 10^{-4}$ mol L$^{-1}$ according to our offline filter test
(Sect. 2.2), and to get the effective concentrations of Cu ions which can participate in the reaction of
the destruction of peroxy radicals, the activity coefficient for total Cu was assumed to be 0.1 (upper
limit) based on a study of $(NH_4)_2SO_4$ particles at 68% RH (Ross and Noone, 1991; Robinson and Stokes,
1970). Using copper ions as a surrogate metal for transition metal ions (TMIs), the potential $HO_2$ loss
onto aqueous ambient aerosols via mechanisms involving TMIs was estimated as (Hanson et al., 1994):
$$\frac{1}{\gamma^{TMI}} = \frac{1}{\alpha^{HO_2}} + \frac{\omega}{H_{eff}RT\sqrt{k^I D_{aq}Q\prime}} \tag{3}$$
where $\alpha^{HO_2}$ is the mass accommodation coefficient of $HO_2$, $\omega$ is the mean $HO_2$ molecular speed (cm
$s^{-1}$), $H_{eff}$ is the effective Henry's Law coefficient, R is the gas constant (J $K^{-1}$ $mol^{-1}$), and $T$ the
temperature (K). $k^I$ is the pseudo-first-order rate constant equal to $k^{II}_{TMI}[TMI]$, where $k^{II}_{TMI}$ is the second
order rate constant for aqueous phase reaction with $HO_2/O_2^-$ and TMI. $Q'$ accounts for aqueous-phase
diffusion limitations and is expressed as
$$Q' = [\coth(q) - \frac{1}{q}]; q = r_p\sqrt{\frac{k^I}{D_{aq}}} \qquad (4)$$
Table S1 shows more details of the parameters used for modeling. Previous laboratory studies suggest
the mass accommodation coefficient for various single-component aerosols dopped with Cu(II) is
commonly > 0.2 (Taketani et al., 2008, 2009; Mozurkewich et al., 1987; Thornton and Abbatt, 2005;
George et al., 2013; Lakey et al., 2016a, 2016b), and organics substantially reduce $HO_2$ uptake onto
aerosols containing TMI (Lakey et al., 2016b). Here we calculated $\gamma^{TMI}$ with $\alpha^{HO_2}=0.2$ using Eq. 3,
which are plotted in Fig. 4a along with the measured $\gamma$ values; the much lower variation of the modeled
values may due to the low time resolution (~2 days) of [Cu]. The measured $\gamma$ values (~ 0.23 on average)
are significantly higher than the modeled $\gamma^{TMI}$ with $\alpha^{HO_2}=0.2$ ( ~ 0.16 on average), with calculated
$p<0.05$ (t-test), which may due to the TMI contained in the ambient aerosol. However, when using the
upper limit of the mass accommodation value for modelling (with $\alpha^{HO_2}=1$), the measured $\gamma$ values are
significantly lower than the modelled $\gamma^{TMI}$ (averaged value: ~ 0.50), these results indicating that the
chemical components may be internally mixed, as proposed by Takami et al. (2013), which influences
the aerosol surface tension and the activity of the free form of the copper ion (i.e., OA and BC) to
constrain $\gamma^{TMI}$. We suggest that the additional collective effects of different chemical components to
$\alpha^{HO_2}$ and the bulk reactions should be involved in the $\gamma^{TMI}$ modelling to get accurate estimation. No
linear correlation was found between $\gamma^{TMI}$ and $\gamma$. Further classification of measured $\gamma \geq 0.4$ and $\gamma < 0.4$
shows that $\gamma^{TMI}$ has a weak correlation with measured $\gamma$ values when $\gamma \geq 0.4$ (Fig. S7), which may due
to the higher fraction of metals in the total mass at measured $\gamma \geq 0.4$ (~12%) than at < 0.4 (~7%);
therefore, the impact of the other chemical components is much lower. The $\gamma$ values obtained here are
comparable with those in previous ambient aerosol studies (Taketani et al., 2008; Zhou et al., 2019b) (Fig.

5b). When compare with single-compound aerosols obtained from laboratory studies, $\gamma$ values were generally higher than the $HO_2$ uptake coefficients onto organic species (Lakey et al., 2015), soot particles (Bedjanian et al., 2005), and the dry state of inorganic aerosols (i.e., $(NH_4)_2SO_4$, NaCl, and $H_2SO_4$), but comparable or lower than aqueous and copper-doped aqueous phases of inorganic species (Fig. 4b) (George et al., 2013; Lakey et al., 2016b; Taketani et al., 2008; Thornton and Abbatt, 2005). This may indicate the collective effects of the individual chemical components of ambient aerosols to $\gamma$, and the significant influence of RH to aerosol states of single-component particles thus their $HO_2$ uptake coefficients.

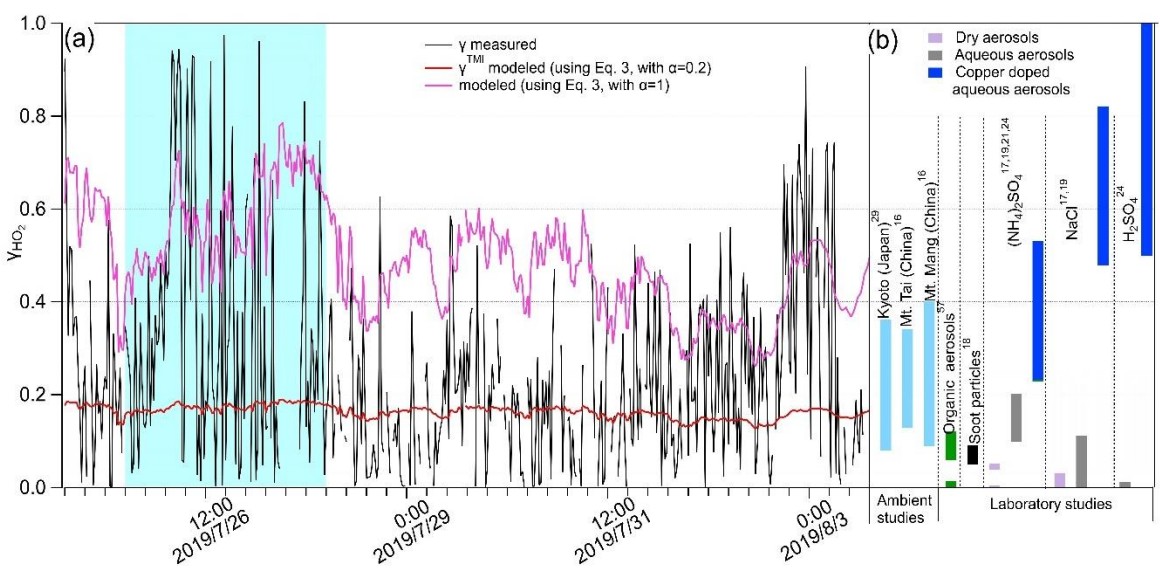

Figure 4: (a) Measured and modeled $\gamma$ values along with measurement time. The blue shaded area represents group i; the remaining areas represent group ii. (b) $HO_2$ uptake coefficients onto different types of aerosol obtained from ambient and laboratory studies, the numbers indicate the related references from which the data were obtained: 1. Zhou et al., 2019b; 2. Taketani et al., 2012; 3. Lakey et al, 2015; 4. Bedjanian et al., 2005; 5. Taketani et al., 2008; 6. George et al., 2013; 7. Lakey et al., 2016b; 8. Thornton and Abbatt, 2005.

Other studies have shown that $\gamma$ is strongly negatively temperature dependent (Remorov et al., 2002; Mao et al., 2010; Cooper and Abbatt, 1996; Hanson et al., 1992; Thornton and Abbatt, 2005; Gershenzon et al., 1995). Here, RH and $T$ were stabilized by the VACES–LFP–LIF system (in the reaction cell), as compared with those in ambient air (Fig. S8), statistical analysis indicates the RH and $T$ in the reaction cell were on average decreased 3.3% (T-test, p<0.05, with inspection level = 0.05) and 2.3 °C (T-test, p<0.05, with inspection level = 0.05), compared to that in ambient air, respectively, which is much smaller than the standard deviation of $T$ and RH (which are ~3.7 °C and 16.4%, respectively) along with the measurement time. We noticed that $k_a$ and $\gamma$ showed no dependence on RH and $T$ in the

reaction cell (see Fig. S9), indicating that the instantaneous change of RH and $T$ may not be the
dominating factors in terms of the variation of $k_a$ and $\gamma$ with measurement time, and the stabilization of
RH and $T$ by VACES–LFP–LIF system have negligible effects to $k_a$ and $\gamma$. This suggests that the
individual chemical components and physical properties of ambient aerosols may dominate $\gamma$ variation
during field campaign; both the metal-catalyzed reactions and the chemical components and their states
should be considered to yield more accurate $\gamma$ values. Results obtained here are in accordance with
previous results on correlations between particulate $H_2O_2$ (which can be formed by the uptake of $HO_2$)
and coarse particulate transition metals (Wang et al., 2010). Using an offline methodology to investigate
the influence of RH and $T$ by maintaining constant experimental conditions or chemical compositions
will be the subject of future work.
**3.5 Influence of the physical properties of ambient aerosols on $k_a$ and $\gamma$**
$HO_2$ heterogeneous loss rates are driven by the different particle sizes of different aerosol types (i.e.,
urban ambient aerosols and marine aerosols)(Morita et al., 2004; Guo et al., 2019; Jacob, 2000). In this
study, $k_a$ and $\gamma$ showed no linear dependence on the mean ambient particle diameters (see Fig. S10).
Identifying the fractional contributions of aerosols in different particle size ranges to $k_a$ and $\gamma$ is highly
desirable in terms of understanding their influence. However, it seems that high $\gamma$ values ($> 0.8$) occur
when the surface area is $< 2\times 10^{-6}$ cm$^2$ cm$^{-3}$ and the mean particle diameter is $< 110$ nm. This is in
accordance with a previous study showing that aerosols yield the highest fractional contribution to the
total heterogeneous loss rate of $HO_2$ radicals of size $< 0.1$ µm (Morita et al., 2004) and that the mass
accommodation process plays the determining role for small and medium sized aerosols in controlling
$HO_2$ uptake. Guo et al. (2019) states the $HO_2$ radicals experience less loss upon its diffusion into larger
droplets than its diffusion into small droplets due to dilution effects make the larger aerosols having
lower depleting species concentrations ($Cu^{2+}$). However, this was based on the assumption that the total
mass of $Cu^{2+}$ is constant during the hygroscopic growth of particles which is not always true in the
ambient conditions. Further studies about $Cu^{2+}$ content in particles with different sizes are needed to
fully understand the result here.
**3.6 Significance of $k_a$ to $O_3$ formation potential**
In urban atmosphere, $XO_2$ (=$HO_2$+$RO_2$) fate is important to the photochemical production of ozone
($P(O_3)$). Here, the loss rates of $XO_2$ due to three factors were compared: (i) uptake onto the ambient
aerosols ($L_{P-XO_2}$ in Eq. 5), since no experiment or reference available for $RO_2$ uptake onto ambient
particles, we assume the $RO_2$ reactivities caused by its interaction with ambient aerosols were the same
as $k_a$, (ii) $XO_2$ self-reactions ($L_{R-XO_2}$ in Eq. 6), and (iii) reaction with NO ($L_{N-XO_2}$ in Eq. 7), which can
produce $NO_2$, a precursor of $O_3$; therefore Eq. 7 can also be regarded as $P(O_3)$.
$$L_{P-XO_2} = k_a[XO_2] \tag{5}$$
$$L_{R-XO_2} = 2 * (k_{HO_2-HO_2}[HO_2]^2 + k_{HO_2-RO_2}[HO_2][RO_2]) \tag{6}$$
$$L_{NO-XO_2} = k_{NO-XO_2}[NO][XO_2] = P(O_3) \tag{7}$$
where $k_{HO_2-HO_2}$ and $k_{HO_2-RO_2}$ are the second-order rate constants of $HO_2$ self-reaction and its reaction
with $RO_2$, respectively. $k_{NO-HO_2}$ is the second-order rate constant of the reaction of $HO_2$ with NO. The
$HO_2$ concentration was estimated from $O_3$ concentration using the method described by Kanaya et al.,
(2007a). The $RO_2$ concentration is then estimated by assuming a steady state of $HO_2$ in the $HO_X$ cycle;
the reaction rates of $HO_2$ radicals are approximated as 0:
$$\frac{d[HO_2]}{dt} = P_{HO_2} - L_{HO_2} = k_{CO-OH}[OH][CO] + k_{H_2CO-OH}[OH][H_2CO] + k_{NO-RO_2}[RO_2][NO] -$$
$$2k_{HO_2-HO_2}[HO_2][HO_2] - k_{HO_2-RO_2}[HO_2][RO_2] - k_{NO-HO_2}[HO_2][NO] - k_a[HO_2] = 0 \tag{8}$$
where $k_{CO-OH}$ and $k_{H_2CO-OH}$ are the second-order rate constants of the reactions of CO and $H_2CO$ with
OH, respectively. The different $XO_2$ loss rates described in Eqs. 5–7, along with the measurement times,
are shown in Fig. 5a. Generally, $L_{P-XO_2}$ is much greater than $L_{R-XO_2}$, indicating that the $XO_2$ taken up
by ambient aerosols will compete with the $XO_2$ self-reaction, thus influencing $XO_2$ concentration.
However, such an influence may have a negligible impact on $P(O_3)$ because $L_{P-XO_2}$ is tens of thousands
of times lower than $L_{NO-XO_2}$ owing to the relatively high NOx concentration at Yokohama. We further
tested the impact of $L_{P-XO_2}$ on ozone formation sensitivity regime, according to the method proposed
by Sakamoto et al. (2019), in which $L_N/Q$ is used as a new indicator:
$$\frac{L_N}{Q} = \frac{1}{1+(\frac{(2k_R[XO_2]+k_{a'})k_{OH-VOCs}[VOCs]}{(1-\alpha')k_{NO-HO_2}[NO]k_{OH-NO_2}[NO_2]})} \tag{9}$$
where $k_{\mathrm{OH-VOCs}}$ and $k_{\mathrm{OH-NO_2}}$ are the second-order rate constants of the reactions of OH with VOCs
and NO$_2$, respectively. $k_{\mathrm{NO-HO_2}}$ is the second-order rate constant of the reaction of NO with HO$_2$. α'
is the proportion of RO$_2$ in XO$_2$. L$_N$ is the OH radical loss rate through its reaction with NO$_2$. (=
$k_{\mathrm{OH-NO_2}}[\mathrm{OH}][\mathrm{NO_2}]$), and Q is the total loss of the HOx radicals in the HOx cycle reaction (= $L_N$ +
$L_{\mathrm{P-XO_2}} + L_{\mathrm{R-XO_2}}$). The regime transition point can be expressed as
$$\frac{L_N}{Q}_{\text{transition}} = (1-\chi)\frac{1}{2} + \chi\frac{1}{3} \tag{10}$$
where $\chi = L_{\mathrm{P-XO_2}}/(L_{\mathrm{P-XO_2}} + L_{\mathrm{R-XO_2}})$. The results indicate that both L$_N$/Q and L$_N$/Q_without_aerosol
(calculated with and without including $k_a'$ in Eq. 9, respectively) were higher than L$_N$/Q_transition,
indicating that ozone formation was VOC-sensitive throughout the campaign and that the aerosol uptake
of XO$_2$ ($k$a') showed no impact on the O$_3$ formation regime (see Fig. 5, here we only consider the
daytime, when photochemical reactions occur). The plots of L$_N$/Q and L$_N$/Q_without_aerosol as a
function of NO concentration show the values were closer to L$_N$/Q_transition (~0.4) at lower NO
concentrations (Fig. S11); therefore, aerosol uptake may play a more important role in the O$_3$ formation
regime at NO levels lower than those observed in this study. The temporal variations in key factors used
in this section are shown in Fig. S12.








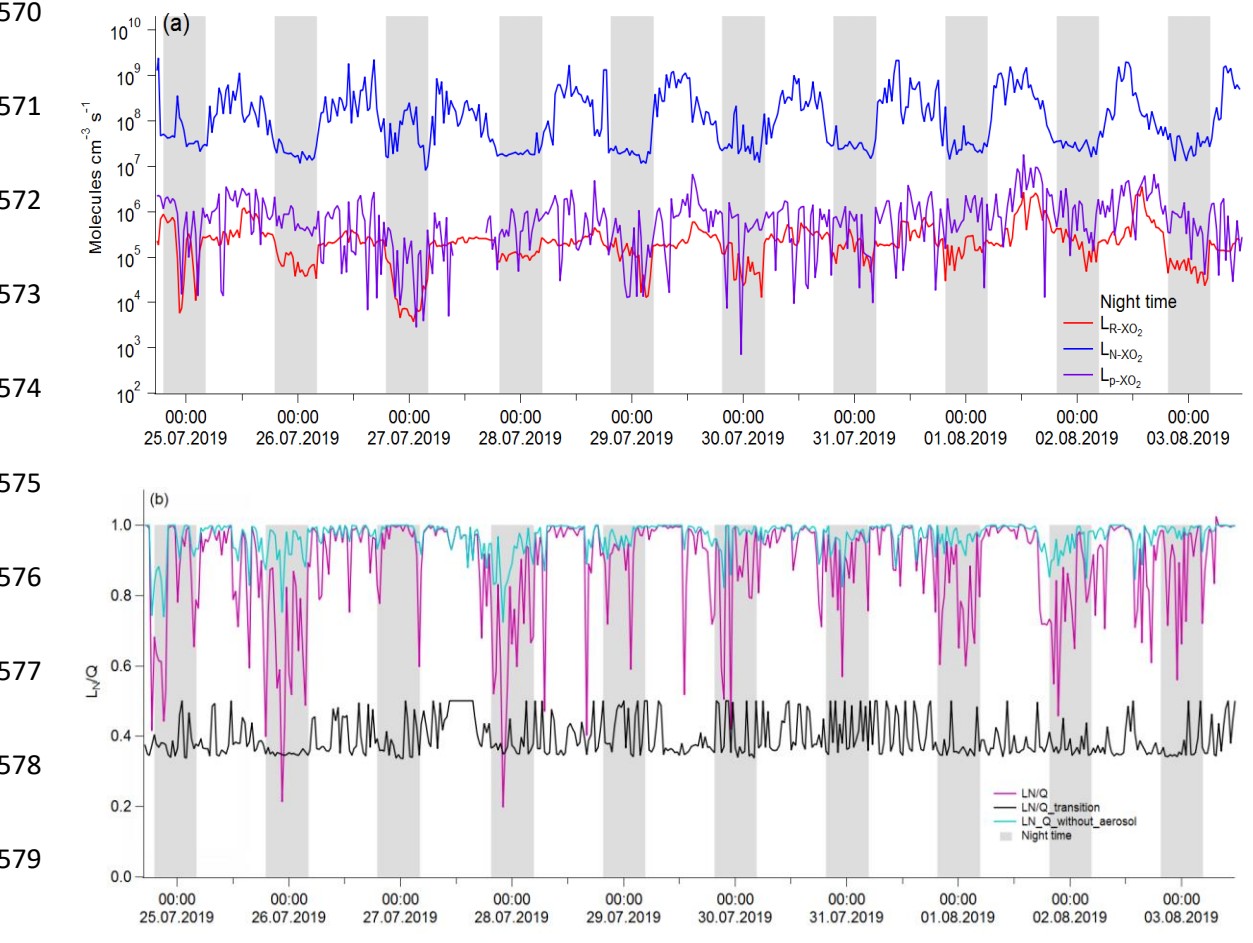

Figure 5: Temporal variations in (a) $HO_2$ radical loss rates and (b) $L_N/Q$ (red line) and the regime transition threshold ($L_N/Q\_transition$, black line) used to assess the ozone sensitivity regime. The gray shaded areas represent nighttime (from National Astronomical Observatory of Japan) and are not discussed herein.

# 4 Conclusions

This study used a reliable online methodology to investigate $HO_2$ uptake kinetics onto in situ ambient aerosols (i.e., $HO_2$ reactivity of ambient aerosols $k_a$ and $HO_2$ uptake coefficients $\gamma$) and discussed the impacting factors on such processes, i.e., chemical compositions and physical properties of ambient aerosols and experimental conditions. $k_a$ ranged between 0.001 s$^{-1}$ (25$^{th}$ percentile) and 0.005 s$^{-1}$ (75$^{th}$ percentile), with an average value of 0.005 ± 0.005 s$^{-1}$. The corresponding $\gamma$, ranged from 0.05 (25$^{th}$ percentile) to 0.33 (75$^{th}$ percentile), with the median value of 0.19 and the average value of 0.23 ± 0.21, is comparable with previous measured (~0.24–0.25) (Zhou et al., 2019b; Taketani et al., 2012) and modeled (~0.20) values (Stadtler et al., 2018; Jacob, 2000). However, the $k_a$ and $\gamma$ values obtained here

are considered as the lower limit values for real ambient aerosols, as the coarse particles were not
measured in this study. We noticed that $k_a$ and $\gamma$ showed no dependence on RH and $T$ in the reaction
cell in this study, indicating that the instantaneous change of RH and $T$ may not be dominating factors
in terms of the variation of $k_a$ and $\gamma$ with measurement time, and the large standard deviation of the $\gamma$
values along with the measurement time ($\pm 0.21$, $1\sigma$) may be due to the instantaneously changed
chemical and physical properties of ambient aerosols, a large bias may exist if a constant $\gamma$ value is used
for modeling.
We found that the individual chemical components of ambient aerosols may have collective effects
of to $\gamma$, through the analyses of 1) separating the air masses into two groups, group i from the ocean and
group ii from mainland Japan; 2) the average diurnal patterns; 3) the correlation matrix analysis of each
individual chemical component of ambient aerosol with $k_a$ and $\gamma$; and 4) the modeling studies using
previously proposed mechanisms. All these efforts clearly indicating that the transition metals contained
in ambient aerosols may act as a catalyst, thus accelerating the depletion of $HO_2$, however, they can be
chelated by OA. OA can also cover the aerosol surface and alter the viscosity of ambient aerosols,
thereby decreasing $\gamma$, and that more oxidized organic aerosols tend to be highly viscous thus decrease
$HO_2$ uptake coefficients. Results obtained here are in accordance to previous laboratory and modeling
studies (Mao et al., 2013a; Lakey et al., 2016b; Lakey et al., 2016a; Takami et al., 2013; Thornton et al.,
2008; Hanson et al., 1994). The chemical components of ambient aerosols may be internally mixed, as
proposed by Takami et al. (2013), which influences the aerosol surface tension and the activity of the
free form of the copper ion (i.e., OA and BC) to constrain $\gamma$. In contrast to previous studies saying that
BC may shrink $HO_2$ losses onto ambient aerosols (Saathoff et al., 2001; Macintyre and Evans, 2011;
Bedjanian et al., 2005), we found BC positively correlated with $HO_2$ uptake coefficients (0.18), this
may be owing to BC can provide active sites or be coated by other chemical components thus facilitating
the physical uptake of $HO_2$. Here, we observed higher $\gamma$ values ($> 0.8$) when the mean particle diameter
is $< 110$ nm, identifying the fractional contributions of aerosols in different particle size ranges to $k_a$ and
$\gamma$ is highly desirable in terms of understanding their influence.
In summary, the chemical components and physical properties of ambient aerosols may dominate
$\gamma$ variation during field campaign; to yield more accurate $\gamma$ value, total suspended particles in ambient
air should be measured, and the metal-catalyzed reactions, chemical components, and aerosol states
should be considered. Also, improvements to the time-resolution of metal measurements are needed for
more precise analysis. For more detailed investigation of $HO_2$ uptake mechanisms, an offline
methodology that can maintain constant chemical compositions and experimental conditions (such as
RH and $T$) will be useful. The $HO_2$ loss onto ambient aerosols was identified to have a negligible impact
on the $O_3$ production rate and formation regime owing to the high $NO_X$ concentrations at Yokohama.
This process may play a more important role in $O_3$ formation under low NOx concentration and high
aerosol loading conditions. The results help us to understand the impacts of $HO_2$ uptake kinetics on
chemical transformations in troposphere.

# Appendix:
Air mass directions (Figure S1), measurement strategy (Figure S2), a technique combined laser-flash
photolysis with laser-induced fluorescence (LFP–LIF), the enrichment of the ambient aerosols, $HO_2$
reactivity of ambient air, correction of gas-phase diffusion for $HO_2$ uptake coefficient, $HO_2$ reactivity
of ambient gas phase ($k_g$), examples of $HO_2$ decay profiles (Figure S3), $HO_2$ reactivity calibration with
different $NO_2$ concentrations (Figure S4), diurnal trends in individual chemical components of ambient
aerosols (Figure S5), diurnal trends in $k_a$ and $\gamma$ (Figure S6), correlations between measured and modeled
$\gamma$ (Figure S7), time series of the averaged RH and $T$ in ambient air and the reaction cell (Figure S8),
dependence of $k_a$ and $\gamma$ on RH in reaction cell (Figure S9), dependence of $k_a$ and $\gamma$ on mean particle
diameter (Figure S10), dependence of day time LN/Q and LN/Q_without_aerosol on [NO] (Figure
S11), profiles of key factors determining $XO_2$ loss rates and P($O_3$) sensitivity (Figure S12), summary
of equations and values used for $\gamma$ modeling (Table S1), summary of equations and values used for $XO_2$
(=$HO_2$+$RO_2$) loss and $O_3$ formation sensitivity regime (Table S1).

# Author contribution
J.J., K.M., Y.S., and Y.K. designed the experiments and J.J. and Y.B. carried them out. J.J. did the data
analysis and prepared the manuscript with contributions from all co-authors.

# Competing interests

The authors declare that they have no conflict of interest.

# Data availability

Data supporting this publication are available upon request for the corresponding author
(junzhou@jnu.edu.cn).

# Acknowledgments

This work was supported by the Japan Society for the Promotion of Science (JSPS) KAKENHI Grant
Numbers JP16H06305, JP19H04255. Many thanks to Yokohama Environmental Science Research
Institute utility during the campaign.

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
