# Peer review of "Kinetics and impacting factors of HO2 uptake onto submicron atmospheric aerosols during a 2019 air quality study (AQUAS) in Yokohama, Japan"

_Atmospheric Chemistry and Physics, 2020_

## Author Response (AR1)

We would like to thank the referees for their careful review and the very valuable suggestions, which improved the quality of the paper a lot. We have addressed the comments point-by-point. The reviewers' comments are in black, our answers are reported in red, and the modifications we have made in the manuscript are highlighted in yellow.

Referee #1:

This is a good paper. $HO_2$ uptake to aerosol can be important as a HOx loss process, especially if NO mixing ratios are low and aerosol surface area is high. There have been a number of lab studies of the uptake coefficient of $HO_2$ to specific aerosol types but to validate these uptake coefficients experiments have to be done in the field with ambient aerosol. However, this requires the use of an online $HO_2$ detection technique (e.g. chemical conversion LIF), an online $HO_2$ source, and aerosol surface characterization techniques, i.e. the experiment is not straightforward to complete.

There are very few online measurements of HO2 uptake using ambient aerosol. This paper presents a new generation of the approach, involving the use of an aerosol concentrator. The concentrator enhances the aerosol surface area and so too the HO2 loss rate, making the experimental results more robust. As well, the authors had a large number of simultaneous aerosol composition measurements, both online and filter based. Importantly, they measured metals by ICP-MS. So, all in all, this is a very challenging but well executed study.

The major results are that the uptake coefficient measured in a field campaign in Yokohama is large, roughly 0.2 to 0.3 on average. This is consistent with lab studies if transition metals are involved. In support, a correlation analysis shows positive correlation between the measured kinetics and metals such as copper.

▪ I have only minor comments, and I recommend that the paper is published.

▪ The paper would benefit from a cleaner description of the role of the mass accommodation coefficient. Most of the lab studies have found values of between 0.2 to 0.5 to be lower limits to the true mass accommodation coefficient, and so I think it is unlikely that alpha controls the uptake. This is the reason that in Figure 4 the model with alpha = 0.2 is predicting too low values. Overall, I found the paper was a bit unclear about how to describe the potential impact/role of alpha.

We apologize for the unclear description of the mass accommodation coefficient. We have summarized the results from the

literatures which are describing the mass accommodation coefficients of different single-components, and modified the description on Page 18, line 453-457:

"Previous laboratory studies suggest the mass accommodation coefficient for various single-component aerosols dopped with Cu(II) is commonly > 0.2 (Taketani et al., 2008, 2009; Mozurkewich et al., 1987; Thornton and Abbatt, 2005; George et al., 2013; Lakey et al., 2016a, 2016b), and organics substantially reduce $HO_2$ uptake onto aerosols containing TMI (Lakey et al., 2016b)."

In order to test the effects of mass accommodation coefficients and TMI to $HO_2$ uptake coefficients, we modeled $\gamma^{TMI}$ using the TMI model (Eq. 3), by assuming the lower and upper limit of the mass accommodation values as 0.2 and 1, respectively. The related descriptions are added on Pg 18, line 457-467:

"Here we calculated $\gamma^{TMI}$ with $\alpha^{HO_2}$=0.2 using Eq. 3, which are plotted in Fig. 4a along with the measured $\gamma$ values; the much lower variation of the modeled values may due to the low time resolution (~2 days) of [Cu]. The measured $\gamma$ values (averaged value: ~ 0.23) are significantly higher than the modelled $\gamma^{TMI}$ with $\alpha^{HO_2}$=0.2 (averaged value: ~ 0.16), with calculated $p<0.05$ (t-test), which may due to the TMI contained in the ambient aerosol. However, when using the upper limit of the mass accommodation value for modelling (with $\alpha^{HO_2}$=1), the measured $\gamma$ values are significantly lower than the modelled $\gamma^{TMI}$ (averaged value: ~ 0.50), these results indicating that the chemical components may be internally mixed, as proposed by Takami et al. (2013), which influences the aerosol surface tension and the activity of the free form of the copper ion (i.e., OA and BC) to constrain $\gamma^{TMI}$. We suggest that the additional collective effects of different chemical components to $\alpha^{HO_2}$ and the bulk reactions should be involved in the $\gamma^{TMI}$ modelling to get accurate estimation."

Even though that the modeled $\gamma^{TMI}$ predicting too low (with $\alpha^{HO_2}$=0.2) or two high values (with $\alpha^{HO_2}$=1) compared to the observed $\gamma$ in Fig. 4, our aim is just to show that due to the multiple-component properties of ambient aerosols, the modeled $\gamma^{TMI}$ cannot represent the observed $\gamma$, thus we suggest that the additional collective effects of the different chemical components to the mass accommodation coefficients and the bulk reaction should be involved in the TMI model for more accurate estimation.

- It would be valuable to point out in the main manuscript to those unfamiliar with the VACES system that it operates by condensational growth followed by inertial separation and then drying. Thus, the particles are not in exactly the same state as they were in the ambient atmosphere. In particular, I wonder whether transition metal ions that were not soluble under ambient conditions were solubilized in the VACES? As well, the morphology of the particles (e.g. phase separated or not) may not be the same after exiting the VACES.

Thanks for your valuable suggestions. In order to describe the VACES system more clearly, we made the descriptions of LFP-LIF and VACES system separately, and added more details of the LIF-LFP-VACES system in the main manuscript on page 5-6, line 123-135:

"**VACES** To compensate for the relatively low ambient aerosol concentrations thus the low $k_a$, and the low limit of detection (LOD) for the HO$_2$ reactivity measurement ($\sim$0.003 s$^{-1}$ with 240 decay integrations), a setup with VACES and an auto-switching aerosol filter was used before LFP–LIF. The VACES was built according to Sioutas et al. (1999), the ambient air sample was drawn into a tank (containing ultra-pure water heated up to $\sim$32 $^o$C) of VACES through a PM$_{2.5}$ cyclone at a flow rate of over $\sim$ 100 L min$^{-1}$, where the ambient air steam was saturated and subsequently cooled down in a condenser connected above the tank (with a temperature of $-2\,^o$C). During this process, the water droplets with diameters >2 $\mu$m formed on the collected ambient aerosols, which were then enriched by a virtual impactor (with a 50% cutoff point less than 1 $\mu$m) and dried by passing through a diffusion dryer connected right after the condenser in sequence. The concentration enrichment of the ambient aerosols can be estimated using the total intake flow of VACES and the minor output flow of the virtual impactor that connected to the aerosol instrumentations (more details are given in SI: the enrichment of the ambient aerosols)."

And on page 7-8, line 180-183:

"**HO$_2$ uptake kinetics** After passing through the VACES system, the ambient air was sampled using a three-port valve (Bolt, Flon Industry Co., LTD) and injected into the LFP–LIF system. The valve was switched automatically between two sampling lines, one with the aerosol filter on, and the other one with the aerosol filter off, …"

Unfortunately, we did not measure the aerosol state change caused by VACES during the measurement campaign. However, we found the RH and $T$ in the reaction cell decreased 3.3% and 2.3 $^o$C on average, respectively, compared to that in ambient air. Due to further analysis (as shown in the response of next question), we believe the effects of $T$/RH to the ambient aerosols' states are negligible.

We did not test the possible change of the solubility of transition metal ions before and after injecting the VACES. However, the VACES was built based on Sioutas et al. (1999), and Wang et al. (2013, 2014) claimed that when using the same technique for aerosol collection for the online measurement of copper, equivalent copper concentrations were obtained compared to those measured by inductively coupled plasma mass spectrometer (ICP-MS) for both total and water-soluble components of ambient PM. This indicates that the influence of VACES to the solubility of Cu in ambient PM is negligible. We added the related discussion on page 6, line 135-139:

"Wang et al. (2013, 2014) claimed that when using the same technique as VACES for the online measurement of copper in ambient aerosols, equivalent copper concentrations were

obtained compared to those measured by inductively coupled plasma mass spectrometer (ICP–MS) for both total and water-soluble components, which indicates the impact of VACES system to the solubility of Cu contained in ambient aerosol is negligible."

Furthermore, according to the study of Yu et al.(2014) and Bertram et al. (2011) about liquid-liquid phase separation in atmospherically relevant particles consisting of organic species and inorganic salts, a phase separation relative humidity ranged from 60%-100%, and the oxygen-to-carbon elemental ratio (O:C) of the organic component appears to be the most useful parameter for estimating the liquid–liquid phase separation, which was always observed for the O:C < 0.5 and was never observed for O:C ≥ 0.8. In the aerosol enrichment process of VACES in this study, the RH changed from ~80% (in ambient air) to ~100%(in water tank), and then to 75%(in reaction cell, the ambient aerosol O:C ranged between 0.1 and 0.7, which suggest that the phase separation may have already happened before entering the VACES system. Thus we assume the morphology of the ambient aerosols didn't change during the enrichment process. We added the related discussion on Pg 6, line 139-147:

"Furthermore, previous studies found the liquid-liquid phase separation RH ranged from 60%-100% in atmospherically relevant particles consisting of organic species and inorganic salts (Yu et al., 2014), and the organic component appears to be the most useful parameter for estimating the liquid–liquid phase separation, which was always observed for oxygen-to-carbon elemental ratio (O:C) < 0.5 and was never observed for O:C ≥ 0.8 (Bertram et al., 2011). In this study, the ambient aerosol O:C ranged from 0.1 to 0.7, and the RH changing from ~80% (in ambient air) to >100% (in water tank), and then to ~75%(in reaction cell), suggest that the phase separation may have already happened before entering the VACES system, thus we assume the morphology of the ambient aerosols didn't change during the concentration enrichment process."

▪ What sizes of particles are concentrated by the VACES? Was there a PM2.5 or PM1 impactor or cyclone on the inlet line? Was there any surface area that the SMPS did NOT measure? Were the temperature and RH of the SMPS systems exactly the same as the T/RH in the flow reactor, i.e. were the aerosol surface areas the same between these two regions? What are the uncertainties in the surface area enrichment factors?

We apologize for the unclear descriptions of the size distribution. We used a $PM_{2.5}$ cyclone before injecting the ambient air into the VACES system, therefore the VACES concentrated $PM_{2.5}$ in the ambient air, but SMPS only measures the surface area of ambient particles with diameter < 0.74 μm. For ambient particles with diameter > 0.74 μm, there is no size distribution measurement. The enriched surface area of $PM_{2.5}$ was propagated from the enriched surface area of $PM_{0.74}$ by assuming the surface area are increased in proportional to the mass concentration. The enriched surface area of $PM_{2.5}$

were then estimated by multiplying the ratio of the mass between the $PM_{2.5}$ and the $PM_{0.74}$ (~1.1) by the enriched surface area of $PM_{0.75}$. However, since the larger particles (here referred to particles ranged from 0.74 to 2.5 μm) tend to have lower surface area than the smaller particles (here referred to particles ranged from 0 to 0.74 μm), we consider the calculated enriched surface area of $PM_{2.5}$ as the upper limit value. The uncertainty of the enriched surface area was calculated from the instrument systematic error of SMPS and the uncertainty of the enrichment factor of VACES. We modified the related description in the modified manuscript on Pg5, line 126-128:

"…the ambient air sample was drawn into a tank (containing ultra-pure water heated up to ~32 °C) of VACES through a $PM_{2.5}$ cyclone at a flow rate of over ~ 100 L min$^{-1}$,..."

And pg7, line173-179:

"The enriched surface area of ambient aerosols with aerodynamic diameter < 0.74 μm ($PM_{0.74}$) was calculated from the surface area of ambient aerosol measured by $SMPS_1$ and the enrichment factor. The enriched surface area of $PM_{2.5}$ was then calculated by multiplying the enriched surface area of $PM_{0.74}$ by the mass ratio between $PM_{2.5}$ and $PM_{0.75}$ (~1.1), where we assume the surface area are increased in proportional to the mass concentration. However, as the larger particles (here referred to particles ranged from 0.74 to 2.5 μm) tend to have lower surface area than the smaller particles, we consider the obtained enriched surface area of $PM_{2.5}$ as the upper limit value. More details can be found in SI."

We added the calculate process of the enriched surface area in supporting information on pg ix, line 212-217:

"The surface area of the enriched ambient aerosols with diameters < 2.5μm ($ES$) are calculated from $S_1$ multiply the ratio between mass concentration of $PM_{2.5}$ and $PM_{0.74}$ (calculated as ~1.1), by assuming the total volume and the total surface area of each size distribution bin (ranged from 0 to 2.5 μm) of $PM_{2.5}$ are increased in proportional with mass concentration compared to that of $PM_{0.75}$ (ranged from 0 to 0.75μm):

$$ES = \frac{PM_{2.5}}{PM_{0.74}} \times S_1 \qquad\qquad (S3)"$$

The uncertainties in the surface area enrichment factors were calculated as the standard deviation of the averaged enrichment factor from 6 days' measurements, using the ratio of the measured surface area post and before the VACES, which is described now on pg7, line162-164:

"The enrichment factor of VACES for the surface area was estimated as $12.5 \pm 2.5$ from the ratio between $S_2$ and $S_1$, where $S_2$ and $S_1$ are the averaged surface areas measured by $SMPS_2$ and $SMPS_1$ of each day, respectively."

The uncertainties of the surface area are described now on pg8, line205-207:

"The uncertainty of the enriched surface area was estimated from the instrument systematic error of SMPS (~ 8%) and the uncertainty of the enrichment factor ($\pm 2.5$), which are shown in Fig.1b (see SI)."

Furthermore, we found the RH and $T$ in the reaction cell were decreased 3.3% and 2.3 °C on average, respectively, compared to that in ambient air, the effects of $T$/RH to the morphology of the ambient aerosols are not clear. But

this change is much smaller than the standard deviation of $T$ and RH (which are ~3.7 °C and 16.4%, respectively) along with the measurement time. Due to further analysis, the HO$_2$ uptake kinetics showed no dependence on the $T$/RH changing along with the measurement time, we believe such small changes in $T$/RH have negligible effects to the aerosol surface areas. We added the related details on pg19-20, line 489-497:

"Here, RH and $T$ were stabilized by the VACES–LFP–LIF system (in the reaction cell), as compared with those in ambient air (Fig. S8), statistical analysis indicates the RH and $T$ in the reaction cell were on average decreased 3.3% (T-test, p<0.05, with inspection level = 0.05) and 2.3 °C (T-test, p<0.05, with inspection level = 0.05), compared to that in ambient air, respectively, which is much smaller than the standard deviation of $T$ and RH (which are ~3.7 °C and 16.4%, respectively) along with the measurement time. We noticed that $k_a$ and $\gamma$ showed no dependence on RH and $T$ in the reaction cell (see Fig. S9), indicating that the instantaneous change of RH and $T$ may not be the dominating factors in terms of the variation of $k_a$ and $\gamma$ with measurement time, and the stabilization of RH and $T$ by VACES–LFP–LIF system have negligible effects to $k_a$ and $\gamma$."

Despite the uncertainties exist in this online VACES-LFP-LIF system, it still accounts for the first direct online particle collection for the measurement of HO$_2$ uptake coefficients, which eliminates uncertainties related to the aerosol sample should be dissolved into the solvent and the incomplete sample extraction from filter media used in offline particle collection method.

- For my version of the paper, the quality of Figure 1 was a bit fuzzy.

  We modified the original figure type and resolution of Figure 1.

- Line 254 – The topic of gas phase diffusion limitations was discussed here for micrometer sized particles. I am confused because earlier in the paper, it was stated that gas phase diffusion limitations were negligible (a few percent at most). As well, the SMPS systems used did not scan up to one micrometer.

  We are sorry that we made a mistake. Initially we wanted to explain that the gas-phase diffusion maybe one reason for the differential of the significantly higher $k_a$ at Yokohama compared to that at Kyoto, but this can be excluded in this study, as we already confirmed the gas phase diffusion limitations can be neglected. We changed the discussion on Pg 13, line 333-335:

  "2) particle size distribution, smaller particles tend to yield higher $\gamma$ values than larger particles owing to the depleting species (e.g., transition metal ions) are mostly distributed in accumulation mode of aerosol,…"

- Line 349 – I don't understand the relationship of the activity coefficient of copper to the topic being discussed.

  We apologize for the unclear description. As the trace elements

including Cu were measured using a Thermo Fisher X2 Series inductively coupled plasma mass spectrometer (ICP–MS) here, the determined concentration is the total metal concentration. Therefore, the effective concentrations of Cu ions (present in free forms) which can participate in the reaction of the destruction of peroxy radicals are highly depend on the relative humidity and aerosol state. In this study, the free form for Cu was calculated using the mean activity coefficients (to be less than 1) tabulated in Robinson and Stokes (1970), as described also in Ross and Noone (1991). To describe this more clearly, we modified the description in the manuscript and also added one more reference on Page 17, line 441-444:

"…and to get the effective concentrations of Cu ions which can participate in the reaction of the destruction of peroxy radicals, the activity coefficient for total Cu was assumed to be 0.1 (upper limit) based on a study of $(NH_4)_2SO_4$ particles at 68% RH (Ross and Noone, 1991; Robinson and Stokes, 1970)."

and the new reference in the reference list:

Robinson, R. A., Stokes, R. H., and Marsh, K. N.: Activity coefficients in the ternary system: water + sucrose + sodium chloride, J. Chem. Thermodyn.,2,745-750,10.1016/0021-9614(70)90050-9, 1970.

▪ The role of BC as a site for HO2 reactivity is tenuous. I agree there is a weak correlation with BC, but I expect the BC will be coated with other aerosol components, and so I don't see the need to invoke reactivity with solely the BC component. Perhaps BC is just a surrogate for some other component of the aerosol, such as a metal?

We think the reviewer invoke a valid point. We have made further investigations on this issue, and believe that both conditions, including BC provide active sites for the $HO_2$ reactions or other aerosol component can coat on the surface of BC thus makes BC as its surrogate for $HO_2$ reaction may happen. We add now the related discussion of the possible conditions on Pg15, line 365-369:

"This may due to the much higher fraction of BC in group i (vs. group ii) may provide active sites for $HO_2$ self-reaction or its reaction with the H atom from the abstraction reaction from hydrogen containing functional groups and form $H_2O_2$ (Bedjanian et al., 2005), or BC can be coated with additional materials (e.g., sulfate and organic carbon), thus influence $HO_2$ uptake (Schwarz et al., 2008)."

and on Pg16, line 412-415:

"According to previous studies, metals may act as a catalyst thus accelerating the depletion of $HO_2$ (Mao et al., 2013a), and BC can provide active sites or can be coated by other chemical components thus facilitating the $HO_2$ uptake (Bedjanian et al., 2005; Schwarz et al. (2008), as described in Sect.3.2. "

and on Pg 17, line 425-426:

"…whereas BC may provide active sites ==or can be coated by other chemical components and thereby increase $\gamma$.== "

and on Pg 24, line 617:

"we found BC positively correlated with $HO_2$ uptake coefficients (0.18), this may be owing to BC can provide active sites ==or be coated by other chemical components== thus facilitating the physical uptake of $HO_2$. "

- Line 72 – I don't understand the comment about electron transfer from O2- to HO2.

    We apologize for this confusing description. We added the reaction process in the description on Pg 3, line 71-72:

"…the uptake of $HO_2$ by ambient aerosols is believed to occur *via* the acid–base dissociation of $HO_2$ ==$(HO_2(g) \leftrightarrow HO_2(aq); HO_2 \leftrightarrow O_2^- + H^+,$== *pKa* = 4.7), followed by electron transfer from $O_2^-$ to $HO_2$ (aq) ==$(HO_2 + O_2^- \xrightarrow{H_2O} H_2O_2 + O_2 + OH^-)$==,…"

- Line 214 – What is the meaning of this "alpha" term?

    This "alpha" term means the inspection level of t-test, with *p*=0.49 (>> 0.05) means the modeled $k_g$ values are not statistically different with the measured $k_g$ values. We apologize for this confusing description, as it is repeated with the mass accommodation coefficients ($\alpha^{HO_2}$), we changed "alpha" to "inspection level" in all related description on Page 11, line 269 and line 272.

- The English in the paper could be improved at places.

    We have sent the revised manuscript again to Enago (www.enago.jp) for proofreading.

References

Bertram, A. K., Martin, S. T., Hanna, S. J., Smith, M. L., Bodsworth, A., Chen, Q., Kuwata, M., Liu, A., You, Y., and Zorn, S. R.: Predicting the relative humidities of liquid-liquid phase separation, efflorescence, and deliquescence of mixed particles of ammonium sulfate, organic material, and water using the organic-to-sulfate mass ratio of the particle and the oxygen-to-carbon elemental ratio of the organic component, Atmos. Chem. Phys., 11, 10995–11006, 10.5194/acp-11-10995-2011, 2011.

Wang, D., Pakbin, P., Saffari, A., Shafer, M. M., Schauer, J. J., and Sioutas, C.: Development and evaluation of a High-Volume Aerosol-into-Liquid Collector for fine and ultrafine particulate matter, Aerosol Sci. Technol., 47, 1226-1238, 10.1080/02786826.2013.830693, 2013.

Wang, D., Shafer, M. M., Schauer, J. J., and Sioutas, C.: Development of a technology for online measurement of total and water-soluble copper (Cu) in $PM_{2.5}$, Aerosol Sci. Technol., 48, 864-874, 10.1080/02786826.2014.937478, 2014.

You, Y., Smith, M. L., Song, M., Martin, S. T., and Bertram, A. K.: Liquid–liquid phase separation in atmospherically relevant particles consisting of organic species and inorganic salts, Int. Rev. Phys. Chem., 33, 43-77, 10.1080/0144235X.2014.890786, 2014.

Referee #2:

The authors present in situ measurements of $HO_2$ reactivity, using an aerosol concentrator and filter to assess specifically the contribution of $HO_2$ uptake to aerosol particles. The authors use simultaneous observations of aerosol composition and surface area to derive an $HO_2$ reaction probability and assess drivers of the variability thereof. This is a nice experiment, and the results are well described and placed in context of previous work. I recommend publication after the authors have addressed some minor comments below.

1) Details on the VACES: The authors should more clearly describe in the main manuscript (not just the SI) the size distribution of aerosol in the reactor as determined by the VACES. Provide mean radius, geometric standard deviation, and ideally compare composition of post VACES aerosol to ambient aerosol measurements. The issue is to what extent is the VACES alterning the particle sizes and types in the reactor compared to the ambient.

We apologize for the unclear descriptions of the size distribution. We used a $PM_{2.5}$ cyclone before injecting the ambient air into the VACES system, according to the test from previous study of the VACES system, the particle concentration enrichment occurs without any coagulation, thus there is no distortion of the size distribution of the original ultrafine aerosols (Sioutas et al., 1999). However, here the size distributions of the ambient aerosols before and after VACES were measured by SMPS for particles <0.74 $\mu$m, there is no measurement of the size distribution >0.74 $\mu$m. We added the mean radius and geometric standard deviation (Geo. Std. Dev.) of the ambient aerosols before and after VACES during the time period of the enrichment factor measurements in Table 1. As shown on Pg 6-7, line 164-172:

"According to the test from previous study of the VACES system, there is no distortion of the size distribution of the original ultrafine aerosols as the particle concentration enrichment occurs without any coagulation (Sioutas et al., 1999), here we listed the mean radius and geometric standard deviation (Geo. Std. Dev.) of the ambient aerosols before and after VACES during the enrichment factor measurement periods, as shown in Table 1. We could see that the mean radius before and after VACES are not statistically different within the standard deviation.

Table 1: The averaged Mean radius and Geometric standard deviation before and after VACES during the time period of the enrichment factor measurements

| Experimental time• | Before VACES | | After VACES | |
|---|---|---|---|---|
| | Mean radius (nm) | Geo. Std. Dev. | Mean radius (nm) | Geo. Std. Dev. |
| 2019.7.25 09:03 − 11:03 | 129.47±11.32 | 0.92±0.04 | 133.19±3.37 | 0.92±0.02 |
| 2019.7.26 09:30 − 11:30 | 94.95±14.42 | 0.99±0.09 | 85.09±14.96 | 1.01±0.09 |
| 2019.7.27 10:00 − 12:00 | 85.09±14.96 | 1.01±0.09 | 80.40±21.01 | 1.01±0.07 |
| 2019.7.28 09:30 − 11:30 | 163.62±13.32 | 1.01±0.08 | 164.06±14.40 | 1.04±0.06 |
| 2019.7.29 09:10 − 11:10 | 128.06±6.90 | 0.91±0.02 | 125.07±7.68 | 0.92±0.02 |
| 2019.7.30 09:30 − 11:30 | 111.40±8.21 | 1.01±0.02 | 115.32±6.26 | 1.01±0.03 |

•represent the time period of the enrichment factor measurements;
±represent the standard deviation of the averaged values of mean radius and Geo. Std. Dev..”

Unfortunately, we did not measure the chemical composition after the VACES, thus we are not able to compare the chemical composition of the post VACES aerosols to ambient aerosol. However, Kim et al. (2001) performed the enrichment test using the ambient aerosol fractions including coarse and fine PM and found that the VACES system does not differentially affect the chemical composition of ambient PM during the enrichment process, thus we assume the chemical composition changing due to the enrichment process of the VACES can be neglected. We have modified the related content and discussion on Page 6, lines 147-152:

“Unfortunately, we did not measure the chemical composition after the VACES, thus we are not able to compare the chemical composition of the post VACES aerosols to ambient aerosol. However, previous test using the ambient aerosol fractions including coarse and fine PM indicated that the enrichment process of the VACES system does not differentially affect the chemical composition of ambient PM (Kim et al., 2001), thus we assume the chemical composition changing due to the enrichment process of the VACES can be neglected.”

2) Neglect of gas-phase diffusion corrections to the determination of the reaction probability seems problematic. My recollection is that limitations are significant (greater than 10%) for gamma > 0.1 and particle sizes > 0.5 micron. It is hard to know from what is provided in the main manuscript whether this issue is dealt with adequately.

We apologize for the unclear description. The mean diameter of the ambient particles ranged from 0.1-0.46 μm, with the median value of 0.25 μm. We detected an error in previous conclusion and calculated the gas-phase diffusion again using the method described in SI. The results show the gas-phase diffusion can increase $\gamma$ for ~ 6.6% (on average), the absolute increase of $\gamma$ due to the gas-phase diffusion is 0.03 on

average, which is much smaller than the $\gamma$ uncertainty (~0.21 on average), therefore, we neglected the gas-phase diffusion corrections to the determination of the reaction probability. The related description is added now on Pg11, line 281-285:

"The mean diameter of ambient particles ranged from 0.1 to 0.46 μm (with the median value of 0.25 μm), the gas-phase diffusion effects on $\gamma$ were estimated to be ~ 6.6 % (further details are given in the SI). The absolute increase of $\gamma$ due to the gas-phase diffusion is 0.03 on average, which is negligible compared to $\gamma$ uncertainty (~0.21 on average), therefore, we ignored the gas-phase diffusion effects to $\gamma$."

3) I appreciate the authors providing the 25th and 75th percentile gamma values, but then state the gamma was 0.33 "on average". Was this the mean, or the median? I would suggest given the variability that the median be reported instead of the mean.

Thank you for the appropriate suggestion at this instance. Judging from the large variation of the mean value, we agree that it is more appropriate to use the median value instead of the mean value of $\gamma$, meanwhile, in order to compare the $\gamma$ value obtained here with previous studies (where used average values instead of median values), we also added the average value as a reference. We have changed the related sentence in the manuscript on Page 11, lines 279-280:

"The corresponding $\gamma$, calculated from Eq. 2, ranged from 0.05 (25[th] percentile) to 0.33 (75[th] percentile), with the median value of 0.19 (with an average value of 0.23 ± 0.21)."

And Page 23, line 592:

"…with the median value of 0.19 and the average value of 0.23 ± 0.21,"

Accordingly, we also describe the median $k_a$ value, instead of mean $k_a$ value on Page 11, line 276:

"…with the median value of 0.005 s$^{-1}$ and average value of 0.005 ± 0.005 s$^{-1}$."

4) I would like to see a deeper assessment of uncertainty at low surface areas and small particles. The derived $k_a$ is likely a small number from the difference of two large numbers with uncertainties due to precision and systematic variability given that a filter must be used serially at a different time to determine $k_a$. The trend towards higher gammas with low surface area and small particles is at best more uncertain and possibly somewhat artificial if a) negative ka are excluded from the analysis, or b) a small positive $k_a$ is divided by a smaller surface area, leading to a bigger gamma, but which isn't robust due to uncertainty (instrumental error).

As it is, the measured gamma time series is extremely noisy - noisier than the aerosol mass (surface area) and composition measured by the

AMS. Some discussion of the different variability in these quantities is warranted and possibly provide shading that indicates the absolute error of each measurement.

We appreciate your suggestion. We actually used $Ek_a$ and the corresponding $ES$ (represent the enriched $k_a$ and enriched surface area of ambient aerosol, respectively) to calculate $\gamma$. The $Ek_a$ was calculated from the significant differences between the measured $HO_2$ reactivity of the [gas phase + enriched ambient aerosols] ($Ek_a+k_g$) and the modeled $HO_2$ reactivity of the gas phase ($\approx k_g$), thus we first estimated the error of $Ek_a$ from ($Ek_a+k_g$) and $k_g$, and then estimated the error of $\gamma$ from the propagate error of $Ek_a$ and $ES$. The uncertainty of the enriched surface area was estimated from the instrument systematic error of SMPS (~8%) and the uncertainty of the enrichment factor($\pm$2.5), as shown on Pg 8, line 205-207:

"The uncertainty of the enriched surface area was estimated from the instrument systematic error of SMPS ($\sim$ 8%) and the uncertainty of the enrichment factor ($\pm$2.5), which are shown in Fig.1b (see SI)."

More details of the related discussion of the different variability in these quantities can be found in on Pg 11, line 276-281:

"The error for $Ek_a$ was estimated as $\sim$ 0.05 s$^{-1}$, calculated as the propagated errors from $k_g+Ek_a$ (i.e., the systematic error of the instrument, ~0.05 s$^{-1}$) and the modeled $k_g$ in mode (b) ($\sim$ 0.001 s$^{-1}$). Accordingly, the errors for $k_a$ was estimated as $\sim$ 0.004 s$^{-1}$ (from the obtained error of $Ek_a$ by dividing by the enrichment factor $E$). The corresponding $\gamma$, calculated from Eq. 2, ranged from 0.05 (25th percentile) to 0.33 (75th percentile), with the median value of 0.19 (with an average value of 0.23 $\pm$ 0.21)."

We didn't exclude the negative $k_a$ from the analysis, there are super small negative $Ek_a$ exists. We add now the absolute errors of ($Ek_a+k_g$), modeled $k_g$, $Ek_a$ and $ES$ in Fig. 1 (gray shading areas). We detect an error in calculating the uncertainty of $\gamma$, which is now corrected in Fig. 1b. We modified the figure caption of Fig. 1 accordingly:

"Figure 1: Temporal variation of parameters under different experimental conditions. (a) Without aerosol phase: 1st panel: measured $NO_2$ concentrations (ppb); 2nd panel: measured (red line) and modeled (black line) $k_g$; 3rd panel: fitting residues of modeled $k_g$ values, ranging from −0.04 (25 percentile) to 0.05 (75 percentile), therefore we consider the systematic error of the LFP–LIF instrument to be ~0.05 s$^{-1}$. (b) Gas + aerosol phase: 1st panel: measured total $HO_2$ reactivity ($k_g+Ek_a$) and modeled $k_g$; 2nd panel: $Ek_a$, calculated from the difference between the measured and modeled values from the 1st panel, the gray shadow area represents the uncertainty of $Ek_a$ ($\Delta Ek_a$), propagated from the error of ($k_g+Ek_a$) and modeled $k_g$; 3rd panel: the upper limit surface area of the enriched ambient aerosols ($ES$), the gray shadow area represents the uncertainty of $ES$ ($\Delta ES$), propagated from the systematic errors of the SMPS instrument (~8%), and the uncertainty of the enrichment factor; 4th panel: $\gamma$ calculated from $Ek_a$ and $ES$ according to Eq. 2. The errors for $\gamma$ were propagated from $\Delta Ek_a$ and $\Delta ES$, $\Delta\gamma=\gamma \times \sqrt{\frac{\Delta Eka^2}{Eka} + \frac{\Delta ES^2}{ES}}$. The blue shaded area represents the air masses from group i (from coast), the remainder is from group ii (from mainland)."

5) Given the lack of size information given in the main paper, it was difficult to assess the role of particle composition, particular the role of sea spray as contributors to surface area, but not measured mass composition (the AMS will not measure sea salt). Thus, this could bias gamma's high if sea salt is unmeasured, or the SMPS do not scan high enough, etc.

We apologize for the unclear descriptions of the size distribution. The VACES system enriches particles with diameter < 2.5 μm, the AMS and SMPS measures particles with diameter < 1 μm and < 0.74 μm, respectively. We modified the related description in the modified manuscript on Pg5, line 126-128:

"…the ambient air sample was drawn into a tank (containing ultra-pure water heated up to ~32 $^o$C) of VACES through a PM$_{2.5}$ cyclone at a flow rate of over ~ 100 L min$^{-1}$,…"

As SMPS only measures the surface area of ambient particles with diameter < 0.74 μm (as shown in Sect. 2.2), there is no size distribution measurement for ambient particles with diameter > 0.74 μm. By assuming the surface area are increased in proportional to the mass concentration, the enriched surface area of PM$_{2.5}$ were estimated by multiplying the ratio of the mass between the PM$_{2.5}$ and the PM$_{0.74}$ (~1.1) by the enriched surface area of PM$_{0.75}$. However, since the larger particles (here referred to particles ranged from 0.74 to 2.5 μm) tend to have lower surface area than the smaller particles (here referred to particles ranged from 0 to 0.74 μm), we consider the calculated enriched surface area of PM$_{2.5}$ as the upper limit value. We modified the related description in the modified manuscript on pg7, line 173-179:

"The enriched surface area of ambient aerosols with aerodynamic diameter < 0.74 μm (PM$_{0.74}$) was calculated from the surface area of ambient aerosol measured by SMPS$_1$ and the enrichment factor. The enriched surface area of PM$_{2.5}$ was then calculated by multiplying the enriched surface area of PM$_{0.74}$ by the mass ratio between PM$_{2.5}$ and PM$_{0.75}$ (~1.1), where we assume the surface area are increased in proportional to the mass concentration. However, as the larger particles (here referred to particles ranged from 0.74 to 2.5 μm) tend to have lower surface area than the smaller particles, we consider the obtained enriched surface area of PM$_{2.5}$ as the upper limit value. More details can be found in SI."

According to SMPS data, most "fine-mode" particles have diameters less than 0.74 μm, with the mean diameter ranged from 0.09 μm to 0.47 μm (with the median value of 0.25 μm), therefore, we assume that particles with diameter ranged between 1 μm and 2.5 μm are negligible. As coarse mode particle generally has size of > 2.5 μm, we believe the discussions about the role of different chemical components with diameter < 1 μm (measured by AMS) played in the HO$_2$ uptake kinetics (with aerosol diameter < 2.5 μm) are reasonable. We add the related discussion on Pg15, line 383-389:

"Here we note that the different chemical components were measured using HR–ToF–AMS for ambient aerosols with aerodynamic diameters < 1 μm, while $k_a$ and $\gamma$ were measured using VACES–LFP–LIF system for ambient aerosols with aerodynamic diameters < 2.5 μm, but due to

most "fine-mode" aerosols have the mean diameter ranged from 0.09 μm to 0.47 μm (with the median value of 0.25 μm, measured by SMPS), we assume the chemical components of ambient aerosols with the aerodynamic diameter ranged between 1 μm and 2.5 μm have negligible impact on Pearson correlation matrix result."

Due to VACES measured particles < 2.5 μm, and AMS measured particles < 1 μm, present results do not include the effects of coarse particles to the $HO_2$ uptake kinetics, we partially missed sea spray (with diameters ranged from ~ 0.05 to 10 μm) effects to the $HO_2$ uptake kinetics, which are described now on Pg 15, line 389-391:

"However, present results do not include the effects of coarse particles (with aerodynamic diameters > 2.5 μm) to the $HO_2$ uptake kinetics, and we may partially miss measuring sea spray (with diameters ranged from ~ 0.05 to 10 μm) effects."

However, we do see a relatively higher chloride concentration in group i from coast (vs. group ii from mainland), and a weak correlation between chloride concentration and $\gamma$ (~0.04) in current analysis, which may be relate to the sea salt reaction, as described on Pg 16, 415-418:

"The very weak correlation of $\gamma$ with $f_{Cl^-}$ (0.04) may be related to $Cl^-$ chemistry, for example, $HO_2(g)$ can react with $NaCl(g)$, produce $NaOH$ and $Cl_2(g)$, thus cause a decrease in the $HO_2$ concentration and indirectly effects $\gamma$ (Remorov et al., 2002)."

When $Cl^-$ measured by AMS increased, coarse particles may exist and our results may not represent the real ambient conditions. Consequently, we consider our results as the lower limit of the $HO_2$ uptake kinetics onto real ambient aerosols. The effects of coarse particle and sea salt to the $HO_2$ uptake kinetics will be the subject of our future study. The related discussion are now add on Pg15, line391-394:

"When $Cl^-$ measured by AMS increased, coarse particles may exist and our results may not represent the real ambient conditions. Consequently, we consider our results as the lower limit of the $HO_2$ uptake kinetics onto real ambient aerosols. "

We also modified the related description in abstract:

"We developed an online approach to precisely investigate the lower limit values of (i) the $HO_2$ reactivities of ambient gases and aerosols and (ii) $HO_2$ uptake coefficients onto ambient aerosols ($\gamma$) during 2019 air quality study (AQUAS) in Yokohama,…"

and in the conclusion on Pg 23-24, line 594-596:

"However, the $k_a$ and $\gamma$ values obtained here are considered as the lower limit values for real ambient aerosols, as the coarse particles were not measured in this study."

and Pg 24-25, line 621-623:

"In summary, the chemical components and physical properties of ambient aerosols may dominate $\gamma$ variation during field campaign; to yield more accurate $\gamma$ value, total suspended particles in ambient air should be measured,…"